# Small Drafts, Big Verdict: Information-Intensive Visual Reasoning via Speculation

**Yuhan Liu**[1], **Lianhui Qin**[2,*], **Shengjie Wang**[1,*]
[1]New York University, [2]University of California, San Diego
{yl10379, sw5973}@nyu.edu
{l6qin}@ucsd.edu
[*]Equal advising

## Abstract

Large Vision-Language Models (VLMs) have achieved remarkable progress in multimodal understanding, yet they struggle when reasoning over information-intensive images that densely interleave textual annotations with fine-grained graphical elements. The main challenges lie in precisely localizing critical cues in dense layouts and multi-hop reasoning to integrate dispersed evidence. We propose Speculative Verdict (SV), a training-free framework inspired by speculative decoding that combines multiple lightweight draft experts with a large verdict model. In the draft stage, small VLMs act as draft experts to generate reasoning paths that provide diverse localization candidates; in the verdict stage, a strong VLM synthesizes these paths to produce the final answer, minimizing computational cost while recovering correct answers. To further improve efficiency and accuracy, SV introduces a consensus expert selection mechanism that forwards only high-agreement reasoning paths to the verdict. Empirically, SV achieves consistent gains on challenging information-intensive and high-resolution visual question answering benchmarks, including InfographicVQA, ChartMuseum, ChartQAPro, and HR-Bench 4K. By synthesizing correct insights from multiple partially accurate reasoning paths, SV achieves both error correction and cost-efficiency compared to large proprietary models or training pipelines. Code is available at `https://github.com/Tinaliu0123/speculative-verdict`.

## 1 Introduction

Recent advances in large vision-language models (VLMs) have delivered impressive performance on tasks such as image captioning and general visual question answering (VQA) (Li et al., 2025c; Fu et al., 2024). However, these models encounter challenges in information-intensive images that densely interleave diverse textual annotations (legends, labels, captions) with fine-grained graphical elements (charts, diagrams, plots) across multiple scales and formats (Su et al., 2025b). Addressing this task requires two interdependent capabilities (Figure 1; Ke et al., 2025): (i) comprehensive and precise localization, which involves not only pinpointing the exact positions of critical cues in densely populated layouts but also ensuring that all query-relevant regions are identified; (ii) multi-hop reasoning, which chains visual analysis—encompassing colors, shapes, and spatial relationships—with textual evidence, thereby integrating dispersed cues into a coherent and complete answer. As each reasoning step builds on the accuracy of the previous one, any intermediate error can propagate through the entire chain, making the overall process highly error-sensitive and difficult to correct retrospectively.

Existing work tackles information-intensive visual reasoning with search-based zoom-in pipelines that enlarge local regions for detailed reasoning. Specifically, learning-based methods train reinforcement learning policies to guide zoom operations iteratively (Zheng et al., 2025; Su et al., 2025a; Fan et al., 2025; Zhang et al., 2025b). Enhancing its performance would demand costly fine-grained supervision. Moreover, training-free methods perform cropping based on internal attention or confidence scores (Zhang et al., 2025a; Shen et al., 2024; Wang et al., 2025c). Yet in dense layouts, we find these signals correlate weakly with true relevance, misleading the model into visually similar but

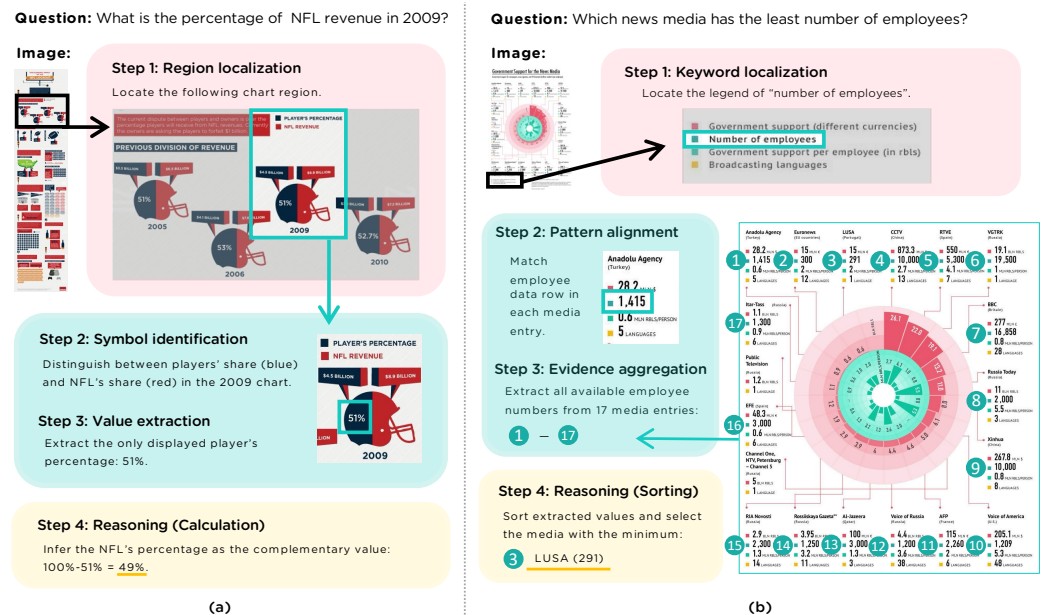

Figure 1: Examples of correct reasoning paths for information-intensive VQA tasks. They illustrate distinct paths: (a) focuses on the localization of a specific chart, symbol identification, and reasoning from a single percentage value; (b) focuses on keyword-based localization, evidence aggregation from multiple entries across the entire image, and cross-entity sorting to select the minimum.

irrelevant areas. Consequently, these tool-driven designs fail to capture all evidence for multi-hop reasoning, leaving the core challenges of information-intensive visual reasoning unsolved.

To overcome these limitations, we propose Speculative Verdict (SV), a training-free framework inspired by speculative decoding that combines small draft visual experts with a large verdict model (Leviathan et al., 2023). The framework operates in two stages (Figure 2): (1) Draft stage: multiple lightweight VLMs serve as draft experts, each generating a reasoning path that offers diverse localization candidates; (2) Verdict stage: a large VLM acts as a strong verdict, which receives the reasoning paths as contextual evidence, distinguishes the correct information, and outputs the final answer. SV directly tackles core challenges through complementary strengths: draft experts expand evidence coverage across scattered regions, while the verdict prevents error propagation by synthesizing these multiple perspectives. Importantly, unlike using a large proprietary model to reason over every image section, SV invokes the verdict only once to yield a concise final answer, thereby minimizing computational cost while effectively recovering correct answers. To further balance accuracy and efficiency, SV introduces a consensus expert selection mechanism in the draft stage, ensuring that only reasoning paths with strong agreement are forwarded to the verdict.

We evaluate SV on information-intensive VQA benchmarks, including InfographicVQA (Mathew et al., 2021), ChartMuseum (Tang et al., 2025), and ChartQAPro (Masry et al., 2025), which demand reasoning over dense textual and visual content. As a training-free framework, SV consistently outperforms strong open-source models, large proprietary models, and perception-focused search methods while remaining cost-efficient. In particular, SV yields average gains of 4% over small VLMs as draft experts and 10% over GPT-4o (Hurst et al., 2024) as verdict. Beyond overall gains, SV successfully corrects 47-53% of cases where majority voting or the verdict model alone fails, thereby reducing vulnerability to error propagation in information-intensive visual reasoning. Furthermore, SV surpasses all baselines on HR-Bench 4K (Wang et al., 2025b), a benchmark for high-resolution visual perception, underscoring its effectiveness in challenging multimodal reasoning scenarios.

## 2 RELATED WORK

**Vision-Language Model Reasoning with Tools.** Recent research has explored enhancing VLM perception by manipulating input images with zooming operations to locate relevant regions. (1) Prompting-based methods exploit internal signals of VLMs to decide where to zoom. ViCrop (Zhang

et al., 2025a) leverages models' attention maps to highlight query-related regions, thereby generating automatic visual crops. Other works perform tree-based search, where models evaluate candidate sub-images with confidence scores to iteratively narrow down to relevant regions (Shen et al., 2024; Wang et al., 2025c). However, such signals align poorly with the required evidence in information-intensive images, since queries often require reasoning across multiple dispersed regions. (2) Reinforcement learning approaches instead optimize policies that interleave visual zooming with textual reasoning (Zheng et al., 2025; Su et al., 2025a; Fan et al., 2025; Zhang et al., 2025b). By calling zooming tools within the agentic framework, these methods adaptively crop regions and concatenate them into the reasoning trajectory, enabling more active evidence gathering. Yet these methods still fall short on information-intensive images, requiring costly task-specific training to scale.

**General Vision-Language Model Reasoning.** Recent work has also explored other paradigms that enhance VLM reasoning. Prompt-enhanced VLMs use chain-of-thought prompting to articulate intermediate observations and sub-goals, yielding more structured reasoning (Xu et al., 2025; Mitra et al., 2024; Shao et al., 2024). RL-enhanced methods further optimize these trajectories via supervised fine-tuning and reinforcement learning, inspired by reasoning-oriented models such as o1 and DeepSeek-R1 (Huang et al., 2025; Ma et al., 2025; Jaech et al., 2024; Guo et al., 2025). Recently, agentic frameworks treat the VLM as a planner that decomposes queries and actively chooses actions, either invoking explicit visual tools or performing implicit latent-space reasoning (Wu & Xie, 2024; Hu et al., 2024; Qi et al., 2024; Yang et al., 2025b; Wu et al., 2025). However, they remain vulnerable to imprecise or weakly supervised visual operations and error propagation.

**LMM-as-a-Judge.** Large multimodal models (LMMs) increasingly serve as general-purpose evaluators for vision-language tasks (Zhang et al., 2023; Ge et al., 2025; Li et al., 2025b). Specifically, LMM judges are prompted or trained to score candidates, produce rankings, or select the best answer given the task context, and instruction (Xiong et al., 2025). These judges can deliver fine-grained evaluations for open-ended generation and reasoning tasks, and are increasingly used as scalable supervision signals for stages such as alignment, retrieval, and reasoning (Li et al., 2025a). In our framework, the verdict model acts as an off-the-shelf multimodal judge that filters informative cues from diverse drafts and synthesizes an answer on information-intensive images.

**Speculative Decoding.** Speculative decoding is a draft-then-verify decoding paradigm to accelerate LLM inference (Xia et al., 2024). Specifically, it uses a draft model to generate future tokens, and a larger target model verifies them via parallel rejection sampling. Recent work extends acceptance from token-level equivalence to step-level semantic similarity to speed up reasoning (Yang et al., 2025a; Pan et al., 2025; Fu et al., 2025b; Liao et al., 2025). Collaborative decoding via Speculation (Fu et al., 2025a) further applies speculative decoding with multiple draft LLMs by verifying proposals against a combined distribution of drafts and target, yielding greater speedups. However, they mainly target speed in LLM inference and do not address visual reasoning challenges.

## 3 SPECULATIVE VERDICT

Speculative decoding is an inference-time optimization originally developed to mitigate the latency of autoregressive generation (Leviathan et al., 2023). The approach employs a draft-then-verify paradigm: (i) a small, fast draft model proposes one or more future tokens speculatively, and (ii) a large, accurate base model verifies these proposals in parallel, accepts or revises the proposals, and generates output that is consistent with the base model's distribution (Xia et al., 2024; Zhang et al., 2024). This token-level process speeds up inference by committing several tokens at once, while maintaining quality by discarding continuations that diverge from the base model's distribution.

The key insight is that draft models expand coverage quickly, while the verifier ensures correctness. Although this idea has been mainly applied to accelerate text generation, its high-level principle is also well-suited for information-intensive multimodal reasoning.

### 3.1 METHOD OVERVIEW

Information-intensive visual question answering (VQA) requires models to localize query-relevant regions, perceive diverse fine-grained textual and visual details, and integrate dispersed evidence into a single correct answer. These tasks are highly error-sensitive as elaborated in Section 1: a single misread or mislocalized element often leads to a completely wrong prediction.

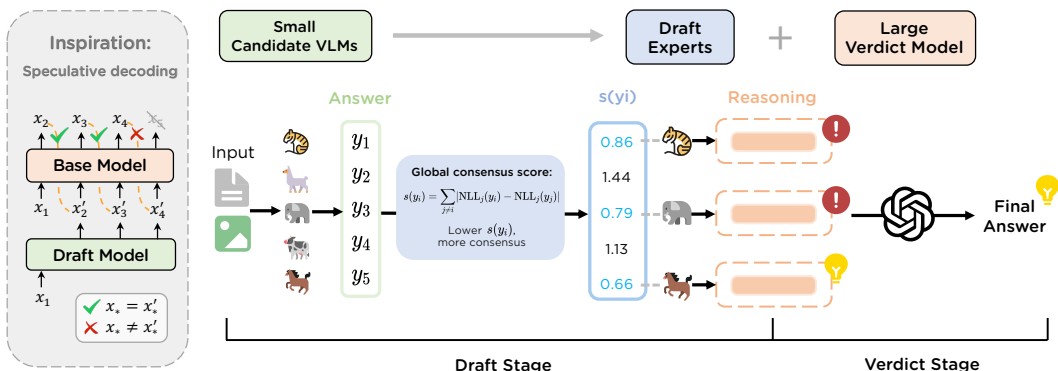

Figure 2: Overview of Speculative Verdict (SV). Inspired by speculative decoding, SV operates in two stages. In the draft stage, given an input question-image pair, $k$ small candidate VLMs first generate candidate answers, from which we compute a global consensus score $s(y_i)$ for each answer based on pairwise NLL differences. We then select $m$ draft experts with the strongest consensus to generate reasoning paths. In the verdict stage, the large verdict model verifies and integrates these paths to yield the final answer.

To address this challenge, we adapt the draft-then-verify paradigm of speculative decoding to multimodal reasoning. Unlike its original use for inference acceleration, we repurpose the paradigm to improve robustness and error correction in information-intensive visual reasoning. On a high level, our Speculative Verdict (SV) framework operates in two stages (Figure 2):

(i) **Draft stage**, where multiple lightweight VLMs are selected as draft experts to provide diverse reasoning paths (Section 3.2);

(ii) **Verdict stage**, where a large VLM acts as verdict to verify, refine, and synthesize these reasoning paths into the final prediction (Section 3.3).

## 3.2 DRAFT STAGE

Chain-of-Thought (CoT) prompting exposes models' intermediate reasoning steps in an explicit, stepwise form (Wei et al., 2022). This is critical for information-intensive VQA, where solving a question requires a sequence of localization, evidence extraction, and analytic operations (Figure 1). However, current VLMs often lack fine-grained perception and localization on densely annotated images, and existing tool-driven zoom-in methods are ineffective as elaborated in Section 2. We therefore utilize multiple VLMs to produce reasoning paths rather than a single direct answer, so that the subsequent verdict can verify and synthesize structured evidence. Concretely, given an image-question pair $(x, q)$, we select $m$ lightweight VLMs $\{M_1, \ldots, M_m\}$ as draft experts from a pool of $k$ candidate VLMs via a consensus-based selection mechanism (detailed in Section 3.4). Each selected expert $M_i$ is then prompted with a CoT template to output a reasoning path $r_i$.

We observe that each reasoning path $r_i$ provided by draft experts typically includes: (i) global scan and localization proposals that identify query-related regions, sections, or subplots, often referencing axes, titles, or captions; (ii) evidence extraction, which transforms visual or textual elements into structured cues, including reading legends, mapping colors to series, parsing axis labels, or assembling lists of values or tokens for subsequent operations; (iii) analytic and reasoning operations, which operate over the extracted cues to derive higher-level conclusions, such as filtering or selecting relevant entities, computing differences, sorting across panels, and cross-referencing dispersed cues. As shown in the running case (Figure 3), different experts may match legends to charts differently; some correctly gather the required cues while others misread adjacent values. This diversity yields a complementary but potentially noisy pool of reasoning signals.

## 3.3 VERDICT STAGE

The set $\{r_i\}$ captures diverse cues, offering richer evidence but also introducing contradictions, which motivates the need for a verdict stage to verify and integrate them. Answer-level ensem-

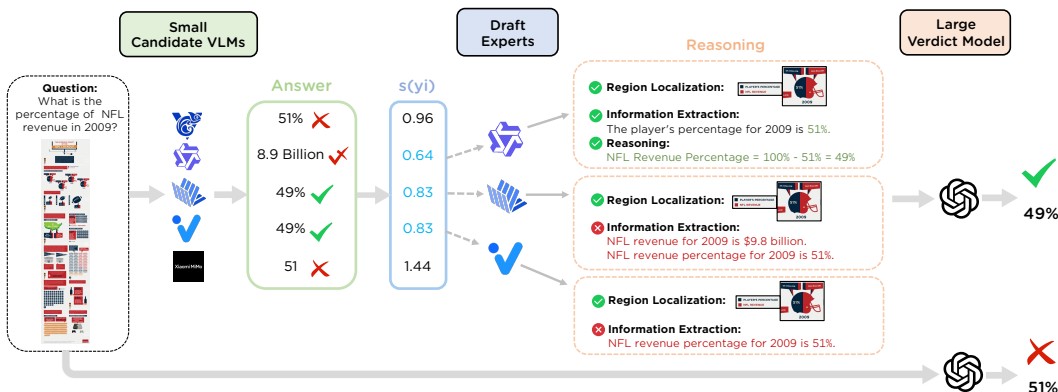

Figure 3: An illustration of Speculative Verdict on InfographicVQA. Five candidate VLMs first produce candidate answers, with only two providing the correct result. Consensus scoring ranks answers by agreement, and the three with the lowest scores are selected as draft experts. Although some experts commit extraction errors (confusing player's share with NFL revenue), the verdict synthesizes their reasoning paths and successfully recovers the correct answer (49%). This illustrates SV's ability to identify reliable experts and achieve error correction.

bling (e.g., majority voting) often fails in minority-correct scenarios where many experts converge on the same incorrect decision, such as mislocalizing the query-related region or misreading fine-grained textual details, even after correct localization. This failure mode is frequently observed in information-intensive reasoning (as illustrated in Figure 3). Rather than discarding minority opinions through majority voting, we leverage a stronger model as a verdict to validate grounding, resolve conflicts, and synthesize coherent reasoning from the draft paths.

Specifically, given the image-question pair $(x, q)$ and the drafts' reasoning paths $\{r_i\}_{i=1}^m$, we prompt the verdict model $J$ with: (i) the original image $x$ as visual input, and (ii) a textual prompt containing the question $q$ and the concatenated reasoning paths $\{r_i\}_{i=1}^m$ as context. The verdict processes this multimodal input in a single inference call and outputs the final answer:

$$y = J\big(x, q, \{r_i\}_{i=1}^m\big).$$

In this design, the verdict acts not as a voter but as a synthesizer. It evaluates grounding consistency, identifies contradictions across reasoning paths, and integrates consistent cues into a coherent prediction. The case in Figure 3 illustrates this intended role: when only one draft extracts the correct evidence, the verdict is designed to recover it by contrasting against competing but inconsistent paths.

This setup enables us to leverage the reasoning capabilities of large models while keeping the inference cost manageable. The verdict stage reduces the expensive autoregressive decoding phase by concentrating computation in prefill: it processes thousands of tokens from multiple draft reasoning paths as prefill input and produces only several answer tokens sequentially. This design avoids invoking large models iteratively for analyzing each image section separately or generating lengthy rationales, both of which would substantially increase decoding costs.

### 3.4 CONSENSUS EXPERT SELECTION

To keep the verdict input both efficient and accurate, we introduce a training-free expert selection mechanism at the beginning of the draft stage (Section 3.2). Since each question in information-intensive VQA has a unique correct answer, consensus among model answers naturally indicates which reasoning paths are more reliable. Therefore, the key idea here is to measure agreement among candidate answers and retain only those with stronger peer consensus. This mechanism is computed efficiently by prefilling the question and answer tokens, with each draft decoded only once, making it plug-and-play with minimal overhead.

**Consensus Score.** We define a consensus score that measures how strongly a candidate VLM's answer is agreed by its peers. Formally, let $x$ be the input image and $q = (q_1, \ldots, q_n)$ the question tokens. From the pool of $k$ candidate VLMs $\{M_i\}_{i=1}^k$, each model produces a candidate answer $y_i = (y_{i,1}, \ldots, y_{i,T})$. For a peer model $M_j$ $(j \neq i)$ in the pool, we measure how plausible it finds $y_i$

by computing the negative log-likelihood (NLL) of the concatenated input $(x, q, y_i)$, i.e., the original image together with the question tokens followed by the candidate answer tokens:

$$\text{NLL}_j(y_i) = -\frac{1}{T} \sum_{t=1}^{T} \log p_{M_j}(y_{i,t} \mid x, q_{\leq n}, y_{i,<t}).$$

To account for calibration differences, we normalize against $M_j$'s own answer $y_j$, thus *relative consensus score* from $M_j$'s perspective is:

$$s_j(y_i) = \left| \text{NLL}_j(y_i) - \text{NLL}_j(y_j) \right|, \quad j \neq i,$$

where a smaller $s_j(y_i)$ indicates stronger agreement, as $M_j$ finds $y_i$ nearly as plausible as its own answer $y_j$.

To capture overall agreement rather than pairwise consistency, we define the *global consensus score* of candidate $y_i$ by summing across all peers:

$$s(y_i) = \sum_{j \neq i} s_j(y_i),$$

which quantifies the overall level of peer consensus for $M_i$'s answer, and a lower $s(y_i)$ indicates stronger agreement and thus higher reliability.

**Consensus Expert Selection Strategy.** We adopt a cross-all strategy that selects the $m$ VLMs with the strongest consensus, measured by the lowest consensus scores, from the pool of $k$ candidates. As described in Section 3.2, the $m$ selected VLMs become the draft experts to generate detailed reasoning paths forwarded to the verdict (Figure 3 illustrates this process). By aggregating agreement across all peers, this strategy provides a holistic measure of reliability. It thus yields a subset of reasoning paths that are well-grounded and compact in size, balancing informativeness and efficiency.

## 4 EXPERIMENTS

### 4.1 SETUPS

**Configuration Details.** We set the draft pool size to $k = 5$ considering efficiency and select $m = 3$ draft experts in our main experiments. Ablation studies over different $m$ values are reported in Section 4.5. The draft pool consists of the following VLMs for expert selection: Qwen2.5-VL-7B-Instruct (Bai et al., 2025), MiMo-VL-7B-RL (Xiaomi, 2025), InternVL3-8B (Zhu et al., 2025), GLM-4.1V-9B-Thinking (Team et al., 2025b), Ovis2.5-9B (Lu et al., 2025). These models are chosen as candidate VLMs based on their strong performance on multimodal benchmarks and their diverse architectural designs. For the verdict models, we employ GPT-4o (Hurst et al., 2024) and Qwen2.5-VL-72B-Instruct respectively, given their superior ability in visual reasoning. In particular, for information-intensive image benchmarks, we preprocess images with PP-StructureV3 (Cui et al., 2025) to produce a layout-preserving structured format, provided together with the original image as auxiliary input to the verdict model.

**Baselines.** We compare SV with proprietary models GPT-4o and GPT-4o-mini, and the large open-source model Qwen2.5-VL-72B-Instruct as it is one of our verdicts. We also evaluate SV against draft experts mentioned above. These baselines are evaluated under the same chain-of-thought prompting template in Appendix M. Additionally, we include DeepEyes (Zheng et al., 2025) and Pixel-Reasoner (Su et al., 2025a) as representative tool-driven baselines with zoom-in operations.

**Benchmarks.** We evaluate SV on three information-intensive benchmarks and extend the evaluation to a representative high-resolution benchmark, providing a comprehensive assessment of fine-grained visual reasoning: InfographicVQA (Mathew et al., 2021), ChartMuseum (Tang et al., 2025), ChartQAPro (Masry et al., 2025) and HR-Bench 4K (Wang et al., 2025b). InfographicVQA collects infographics with an average high resolution over 2k, designed to test reasoning over layout, graphical and textual content, including operations such as counting, sorting, and basic arithmetic. Chart-Museum and ChartQAPro introduce substantially greater visual reasoning complexity by covering a broad spectrum of real-world chart types and question formats, revealing a large performance gap between current Large VLMs and humans. These benchmarks require models to visually ground relevant regions, extract information, and conduct reasoning to answer queries.

Table 1: Results on test sets of four benchmarks. InfographicVQA, ChartMuseum, and ChartQAPro are information-intensive VQA benchmarks, while HR-Bench 4K focuses on high-resolution perception. We compare SV against closed-source, open-source VLMs, and tool-driven methods. † denotes results reported in the original papers and all other results are reproduced by ourselves. The best results for each benchmark are highlighted in **bold** and the second-best results are underlined.

| Model | Param Size | InfographicVQA ANLS | ChartMuseum Acc | ChartQAPro Acc | HR-Bench 4K Acc |
|---|---|---|---|---|---|
| *Closed-source VLMs* | | | | | |
| GPT-4o | – | 76.5 | 42.7 | 52.6 | 67.4 |
| GPT-4o-mini | – | 67.2 | 31.5 | 44.1 | 53.8 |
| *Open-source VLMs* | | | | | |
| Qwen2.5-VL-Instruct | 7B | 79.8 | 29.5 | 51.0 | 73.0 |
| MiMo-VL-RL (think) | 7B | 83.5 | 29.0 | 57.3 | 72.3 |
| InternVL3 | 8B | 72.3 | 25.9 | 45.1 | 68.0 |
| GLM-4.1V-Thinking | 9B | 84.8 | 48.0 | 56.2 | 72.3 |
| Ovis2.5 | 9B | 81.7 | 34.0 | 55.9 | 69.5 |
| Qwen2.5-VL-Instruct | 72B | 84.2 | 40.7 | 60.7 | 73.1 |
| *Tool-driven method* | | | | | |
| DeepEyes | 7B | 75.5 | 28.0 | 48.7 | 73.0 |
| Pixel-Reasoner | 7B | 84.0† | 25.9 | 39.3 | - |
| **SV w/ GPT-4o Verdict** | – | **88.4** | **49.3** | **64.0** | 71.4 |
| Δ (vs. GPT-4o) | – | +11.9 | +6.6 | +11.4 | +4.0 |
| **SV w/ Qwen2.5-VL-72B-Instruct Verdict** | – | 86.7 | 48.2 | 63.0 | **75.6** |
| Δ (vs. Qwen2.5-VL-72B-Instruct) | – | +2.5 | +7.5 | +2.3 | +2.5 |

Table 2: Results on additional multimodal reasoning benchmarks. We evaluate on MathVista-testmini and 1000 randomly sampled complex questions from TallyQA to assess generalization.

| Model | Param Size | TallyQA-Complex Acc | MathVista Acc |
|---|---|---|---|
| *Closed-source VLMs* | | | |
| GPT-4o | – | 75.4 | 65.1 |
| *Open-source VLMs* | | | |
| Qwen2.5-VL-Instruct | 7B | 72.4 | 68.2 |
| MiMo-VL-RL | 7B | 72.0 | 80.3 |
| InternVL3 | 8B | 70.1 | 72.7 |
| GLM-4.1V-Thinking | 9B | 73.9 | 78.9 |
| Ovis2.5 | 9B | 71.9 | 77.3 |
| **SV w/ GPT-4o Verdict** | – | **76.9** | **82.9** |
| Δ (vs. GPT-4o) | – | +1.5 | +17.8 |

We further assess generalization to high-resolution perception on HR-Bench 4K. It comprises two sub-tasks: FSP (Fine-grained Single-instance Perception) and FCP (Fine-grained Cross-instance Perception), stressing small-object perception and cross-instance reasoning. We also test on two additional multimodal reasoning benchmarks, TallyQA (Acharya et al., 2019) and MathVista (Lu et al., 2023), covering open-ended counting with relational reasoning and mathematical visual reasoning.

## 4.2 RESULTS ON INFORMATION-INTENSIVE BENCHMARKS

As shown in Table 1, SV demonstrates superior performance across all benchmarks, outperforming a wide range of baselines. Based on the results, we have the following key observations:

(i) **SV shows consistent gains over all strong draft experts' baselines**, with improvements of 3.6% on InfographicVQA, 1.3% on ChartMuseum, and 6.7% on ChartQAPro with GPT-4o as verdict. SV also achieves comparable gains with Qwen2.5-VL-72B-Instruct as a verdict.

(ii) **Importantly, SV enables strong error correction beyond simple answer aggregation.** Figure 4 analyzes SV's performance on cases where the verdict itself fails, categorized by expert correctness (minority-correct, majority-correct, zero-correct). Across benchmarks, SV recovers 47-53% of minority-correct cases, where few draft experts are correct and the verdict alone also fails (case in Figure 3). Moreover, SV even recovers 2.5-4.5% of zero-correct cases, where neither the drafts nor the verdict answers correctly (case in Appendix L). In these hard cases, SV exploits complementary reasoning strengths across draft experts (e.g., extraction, localization,

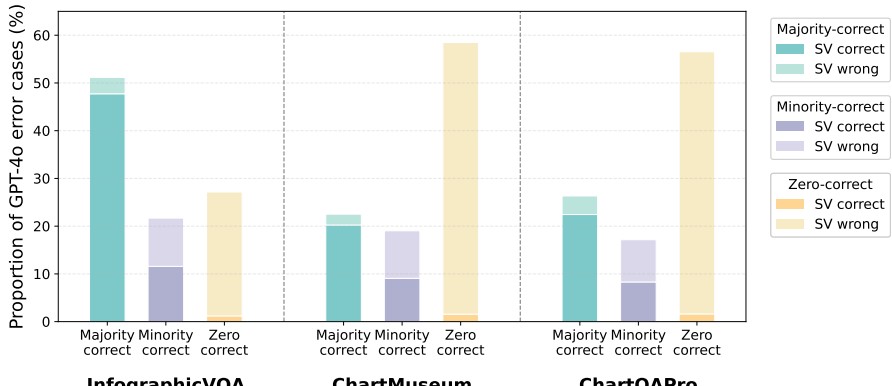

Figure 4: SV's correction ability on verdict's error cases across information-intensive benchmarks (GPT-4o as verdict). We consider only cases where the verdict itself fails, to isolate SV's independent correction capacity. For each benchmark, three bars denote expert correctness categories (majority-correct, minority-correct, and zero-correct), defined by how many selected experts provide the correct answer. Within each category, the bars are split into the proportion corrected by SV (dark) versus not corrected (light). More details can be found in Appendix D.

color matching) (see detailed analysis in Appendix I). This complementarity allows the verdict to synthesize partially correct evidence across reasoning paths while rejecting misleading signals. Thus, SV achieves effective correction where traditional ensemble methods fail.

(iii) **SV strengthens large verdict models significantly**, and using GPT-4o as verdict delivers stronger results due to its reasoning advantage on information-intensive benchmarks. Specifically, when GPT-4o is used as verdict, SV surpasses the GPT-4o baseline by 11.9% on InfographicVQA, 6.6% on ChartMuseum, and 11.4% on ChartQAPro. These improvements come with reduced inference cost for the large verdict model, demonstrating that SV can outperform much larger or proprietary LVLMs in a cost-efficient manner.

(iv) **SV substantially outperforms representative tool-driven pipeline DeepEyes and Pixel-Reasoner.** Specifically, it improves over DeepEyes by 12.9% on InfographicVQA, 21.3% on Chart-Museum, and 15.3% on ChartQAPro and it also exceeds Pixel-Reasoner by clear margins. While these methods benefit from zoom-in operations, such operations are often under-triggered or misdirected on dense infographics (see Appendix K). Thus, it struggles with global comparison and dispersed evidence synthesis. Yet, SV's reasoning-path synthesis enables it to integrate evidence across regions reliably without relying on predefined tool-based visual search.

### 4.3 RESULTS ON HIGH-RESOLUTION BENCHMARK

We further assess generalization to high-resolution images using HR-Bench 4K to evaluate whether SV can enhance fine-grained visual perception. The key observations are as follows (Table 1):

(i) With Qwen2.5-VL-72B-Instruct as verdict, SV achieves its largest margin, surpassing the best-performing draft expert by 2.6% and even outperforming the verdict itself by 2.5%. The superior performance of Qwen2.5-VL-72B as verdict on this task correlates with its stronger visual localization capabilities, indicating verdict selection should align with task-specific requirements.

(ii) SV also exceeds DeepEyes, which is explicitly trained with zoom-in tools for iterative visual search on high-resolution perception. This highlights SV's ability to generalize to high-resolution tasks, where accurate recognition of small objects is critical. Aligning perceptually strong draft experts with a verdict thus provides a simpler yet effective solution for high-resolution reasoning.

### 4.4 RESULTS ON MULTIMODAL REASONING BENCHMARKS

To examine SV's broader generalization, we further evaluate it on two additional multimodal reasoning benchmarks: TallyQA (counting) and MathVista (mathematical reasoning). Table 2 shows that SV provides consistent gains, outperforming the strongest draft expert by 3.0% on TallyQA-

Complex and 2.6% on MathVista, and improving over GPT-4o by 1.5% and 17.8%, respectively. These results demonstrate that SV generalizes to diverse visual reasoning tasks.

## 4.5 ABLATION STUDY

To better understand the effectiveness of SV, we conduct ablation studies on information-intensive benchmarks to analyze the impact of individual components. In these experiments, the reasoning baseline refers to the best-performing draft expert in our pool for each benchmark (Table 1).

**Number of Draft Experts.** Our setting with $m = 3$ draft experts yields a favorable trade-off between accuracy and efficiency, as it determines the number of reasoning paths forwarded to the verdict. As shown in Figure 5, we observe that the performance improves nearly linearly up to three draft experts and then saturates, while inference cost grows roughly linearly with $m$.

**Consensus Expert Selection Strategy.** We confirm the effectiveness of our cross-all selection strategy by comparing it with a best-reference strategy. In the best-reference variant, the top-performing draft expert serves as reference and the two most consistent experts are selected with it. While best-reference is expected to be the strongest criterion, cross-all achieves comparable gains while remaining reference-free (Figure 6).

**Selection Criteria.** Selecting consensus-based experts consistently improves performance, while divergent selection can even fall below the single-draft reasoning baseline (Figure 7). These results support that, for information-intensive tasks, consensus-based selection more reliably identifies the correct reasoning path than enforced diversity.

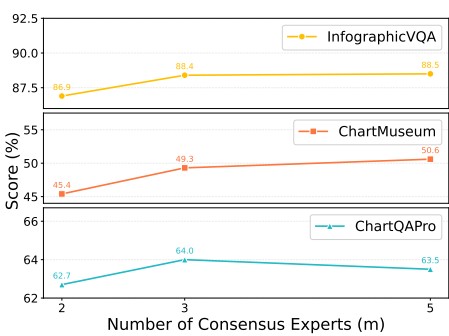

Figure 5: Ablations on the number of draft experts $m$.

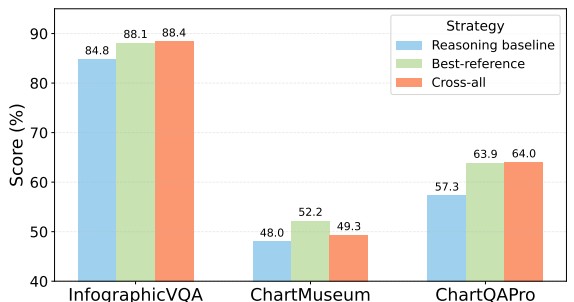

Figure 6: Ablations on different consensus expert selection strategies.

**Impact of Verdict Stage.** The verdict stage yields higher performance than majority voting across information-intensive benchmarks (Figure 8). Notably, majority voting with all five candidate models performs comparably as majority voting with three draft experts, consistent with our finding that consensus selection can match the performance of all drafts at a lower cost (Table 5). SV further surpasses both by leveraging the verdict's error correction ability, successfully capturing minority-correct cases that majority voting discards (Figure 4 and Figure 3).

Beyond majority voting, SV consistently outperforms a strong LMM-as-a-Judge baseline (LLaVA-Critic-72B (Xiong et al., 2025)) by 4.9–11.9% (Figure 8). In this setting, LLaVA-Critic scores or ranks the same draft experts selected by SV and outputs the single best candidate. This advantage is attributed to SV's synthesis-based verdict, which cross-checks and integrates complementary factual cues across multiple trajectories, rather than relying on selecting one trajectory that may be favored by reasoning style. Detailed ablations with LLaVA-Critic are provided in Appendix G.

**Choice of Verdict Textual Input.** Providing full reasoning paths to the verdict yields substantially better performance than passing only final answers (Table 3), with improvements of 15% on InfographicVQA, and 4.8% on ChartQAPro. These results highlight that rich contextual evidence is essential for the verdict to recover correct reasoning, whereas final predictions alone are insufficient.

**Choice of Verdict Scale.** Using a large verdict model yields stronger gains than a small verdict model. We evaluate three strong small verdicts (i.e., GLM-4.1V-9B-Thinking, MiMo-VL-7B-RL, Qwen2.5-VL-7B-Instruct), and all of them underperform SV across benchmarks (Table 4). Notably,

the small reasoning verdicts generate 60-200x more output tokens yet still yield weaker performance, indicating worse cost-efficiency trade-offs (details in Appendix C). The results validate SV's design principle of invoking a strong verdict only once to achieve robust and efficient error correction.

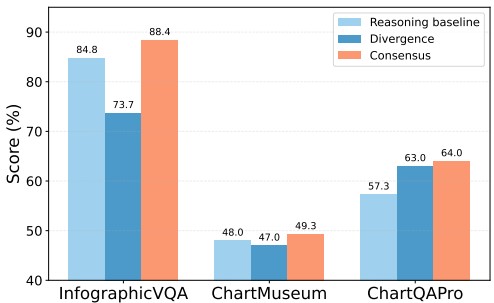

Figure 7: Ablations on selection criteria.

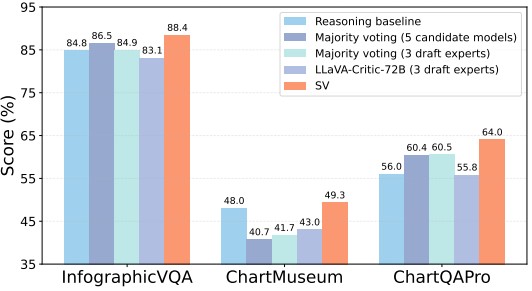

Figure 8: Performance comparison on SV, majority voting and LLaVA-Critic with different model sets.

Table 3: Ablations on verdict textual input.

| Textual input | InfographicVQA ANLS | ChartQAPro Acc |
|---|---|---|
| Reasoning baseline | 84.8 | 57.3 |
| Answers only | 73.4 | 59.2 |
| Reasoning paths (SV) | **88.4** | **64.0** |

Table 4: Ablations on verdict scale.

| Verdict Choice | InfographicVQA ANLS | ChartMuseum Acc | ChartQAPro Acc |
|---|---|---|---|
| Reasoning baseline | 84.8 | 48.0 | 57.3 |
| GLM-4.1V-9B-Thinking Verdict | 84.7 | 48.0 | 59.4 |
| MiMo-VL-RL-7B Verdict | 85.4 | 46.9 | 60.3 |
| Qwen2.5-VL-Instruct 7B Verdict | 84.1 | 47.1 | 57.2 |
| GPT-4o Verdict (SV) | **88.4** | **49.3** | **64.0** |

## 4.6 Cost-Efficiency Analysis

We quantify SV's cost-efficiency by comparing it against large reasoning model baselines, including GPT-4o and o1. As described in Section 3.3, SV achieves cost-efficiency through its verdict stage by concentrating computation in prefill and substantially reducing expensive autoregressive decoding.

As shown in Table 5 and 6, SV consistently improves over GPT-4o by 6.6–12.0% across datasets while maintaining comparable cost. Compared with o1, SV attains substantially better performance on InfographicVQA and ChartQAPro and comparable performance on ChartMuseum while requiring only 15–26% of o1's cost. These results confirm that SV delivers markedly superior cost-efficiency, achieving and even outperforming o1-level performance with lower computational cost.

Table 5: Performance on 1000 randomly sampled test instances per benchmark.

| Method | InfographicVQA | ChartMuseum | ChartQAPro |
|---|---|---|---|
| GPT-4o | 76.3 | 42.7 | 51.7 |
| o1 | 77.8 | **50.6** | 58.8 |
| **SV w/ GPT-4o Verdict** | **88.3** | 49.3 | **63.4** |

Table 6: Average inference API cost per sample across benchmarks.

| Method | InfographicVQA | ChartMuseum | ChartQAPro |
|---|---|---|---|
| GPT-4o | $0.0038 | $0.0213 | $0.0210 |
| o1 | $0.0263 | $0.0663 | $0.0478 |
| **SV w/ GPT-4o Verdict** | **$0.0068** | **$0.0109** | **$0.0071** |

## 5 Conclusion

This paper introduces Speculative Verdict (SV), a training-free framework to address challenges of information-intensive visual reasoning. Inspired by speculative decoding, SV repositions large models as efficient synthesizers rather than computationally expensive step-by-step reasoners. By integrating diverse reasoning paths from lightweight experts, the verdict can distinguish informative cues and recover correctness from structured errors. Experiments show that SV consistently outperforms strong proprietary, open-source, and tool-driven methods, establishing a cost-efficient paradigm for reasoning on information-intensive images.

## 6 ACKNOWLEDGEMENTS

This work was supported in part by Shanghai Frontiers Science Center of Artificial Intelligence and Deep Learning at NYU Shanghai and NYU IT High Performance Computing resources and services. We also thank the anonymous reviewers for their constructive comments and suggestions, which helped improve the quality of this manuscript.

## 7 ETHICS STATEMENT

All authors have read and commit to adhering to the ICLR Code of Ethics. This work does not involve human subjects, sensitive personal data, biometrics, or medical information. All datasets used are publicly available under permissible licenses and are not privacy-sensitive. We recognize that any automated reasoning system may produce incorrect or misleading outputs. To ensure responsible use, we emphasize that our method is intended for research and analysis rather than deployment in high-stakes settings. Users are encouraged to verify model outputs and apply human oversight when necessary. We take full responsibility for all reported results, analyses, and claims, and we welcome community scrutiny and feedback.

## 8 REPRODUCIBILITY STATEMENT

To support reproducibility, we provide comprehensive implementation details throughout our paper. Key experimental configurations, such as draft expert selection, consensus scoring computation, and verdict model specifications, are documented in Section 3.4 and Section 4.1. Detailed prompt templates are presented in Appendix M. In addition, our released code offers concrete implementation details and enables faithful reproduction of all reported results.

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

## A    DATASET STATISTICS

Table 7 reports the statistics of the four major evaluation benchmarks. All benchmarks are based on **real-world images** rather than synthetic renderings, ensuring the authenticity and diversity of the evaluation setting. In particular, InfographicVQA, ChartMuseum, and ChartQAPro are information-intensive benchmarks: they contain thousands of images and questions with dense textual and numerical content, collected from diverse sources spanning 2594, 157, and 184 distinct web domains respectively (Mathew et al., 2021; Tang et al., 2025; Masry et al., 2025). This diversity reduces source bias and reflects practical challenges in multimodal reasoning.

HR-Bench 4K is used primarily to evaluate the generalization of SV, serving as a high-resolution benchmark with average sizes exceeding 4000×3500 (Wang et al., 2025b). Meanwhile, one of our main benchmarks, InfographicVQA, also exhibits high-resolution characteristics. In particular, it contains many images where diagrams span large vertical layouts (see the case in Figure 3), which further compounds the difficulty of grounding and multi-hop reasoning across dispersed regions.

Table 7: Statistics of the evaluation benchmarks. We report the number of images and questions, as well as the average image resolution (width $\bar{W}$ and height $\bar{H}$).

| Dataset | Real vs. Synthetic | #Images | #Questions | $\bar{W}$ | $\bar{H}$ |
|---|---|---|---|---|---|
| InfographicVQA (test) | Real | 3288 | 579 | 1092 | 2771 |
| ChartMuseum (test) | Real | 1000 | 818 | 1551 | 1213 |
| ChartQAPro | Real | 1948 | 1341 | 1194 | 986 |
| HR-Bench 4K | Real | 800 | 200 | 4024 | 3503 |

## B    ADDITIONAL RELATED WORK

**Large Language Model Ensemble.** Majority voting aggregates answers by frequency, but fails when the correct solution is produced by a minority. Universal Self-Consistency (Chen et al., 2023) mitigates this failure mode by prompting the LLM to select the most consistent candidate across samples. Further, learned aggregators read multiple rationales and synthesize them to recover minority-correct information (Qi et al., 2025; Zhao et al., 2025). However, these approaches focus on text-only ensembling. In vision-language reasoning, supervision of ensembling is not cost-effective since multimodal complexity requires costly, fine-grained annotations.

## C    ADDITIONAL COST–EFFICIENCY ANALYSIS

### C.1    AVERAGE API COST OF SV WITH GPT-4O VERDICT

Table 8 reports the average inference cost of invoking GPT-4o as the verdict model per sample across benchmarks. Costs are estimated using the official GPT-4o pricing (version gpt-4o-2024-08-06) as of November 2025. The small variation across benchmarks is mainly attributed to differences in reasoning path length, as more challenging tasks typically induce more complex reasoning. Overall, the inference cost of using GPT-4o as the verdict is under $0.011 per sample across all benchmarks.

Table 8: Average inference cost of GPT-4o as verdict per sample across benchmarks. Costs are computed using GPT-4o (gpt-4o-2024-08-06) pricing by November 2025.

| Dataset | GPT-4o cost per sample |
|---|---|
| InfographicVQA | $0.0068 |
| ChartMuseum | $0.0109 |
| ChartQAPro | $0.0071 |
| HR-Bench 4K | $0.0044 |

## C.2 Average Output Tokens across Verdicts

To compare the token efficiency of replacing the large verdict with smaller models, we keep the draft stage identical and substitute the verdict with different draft models. All verdict models are prompted to output only the final answer, and their performances are reported in Table 4 in the main paper.

As shown in Table 9, the small reasoning models, GLM-4.1V-9B-Thinking and MiMo-VL-7B-RL, generate 60–200× more output tokens than GPT-4o, yet still underperform SV. This highlights SV's advantage: using a strong verdict once for compact verification and synthesis is substantially more token-efficient than relying on smaller verdicts that require long autoregressive reasoning to reach a decision.

Table 9: Average verdict output tokens per sample across benchmarks under different verdicts.

| Verdict model | InfographicVQA | ChartMuseum | ChartQAPro |
|---|---|---|---|
| GLM-4.1V-9B-Thinking | 272.1 | 604.5 | 440.5 |
| MiMo-VL-7B-RL | 183.2 | 368.2 | 378.1 |
| Qwen2.5-VL-7B-Instruct | 3.3 | 3.4 | 2.8 |
| GPT-4o (SV) | 2.7 | 3.0 | 2.2 |

## D Supplementary Recovery Analysis on Information-Intensive Benchmarks

Table 10 and Figure 9 show the detailed recovery statistics across benchmarks with GPT-4o as verdict. We break down SV's performance by expert correctness: (i) cases where the majority of draft experts are correct (majority-correct), (ii) cases where only a minority are correct (minority-correct), (iii) cases where none are correct (zero-correct). While the main paper focuses on the GPT-4o's error cases to isolate SV's effectiveness, we provide the full results here for completeness. Notably, in the zero-correct setting, recovery occurs rarely (2.6-24%), but it demonstrates verdict's surprising ability to infer the correct answer by synthesizing signal from entirely noisy reasoning.

Table 10: Recovery accuracy (%) with GPT-4o as verdict. Results are conditioned on whether GPT-4o itself can produce the correct answer.

| Dataset | GPT-4o Correct | | | GPT-4o Wrong | | |
|---|---|---|---|---|---|---|
| | Majority-correct | Minority-correct | Zero-correct | Majority-correct | Minority-correct | Zero-correct |
| InfographicVQA | 96.81 | 64.13 | 20.54 | 93.30 | 53.42 | 4.44 |
| ChartMuseum | 98.46 | 69.84 | 15.38 | 89.92 | 47.71 | 2.69 |
| ChartQAPro | 94.59 | 68.18 | 24.00 | 85.25 | 48.43 | 2.86 |

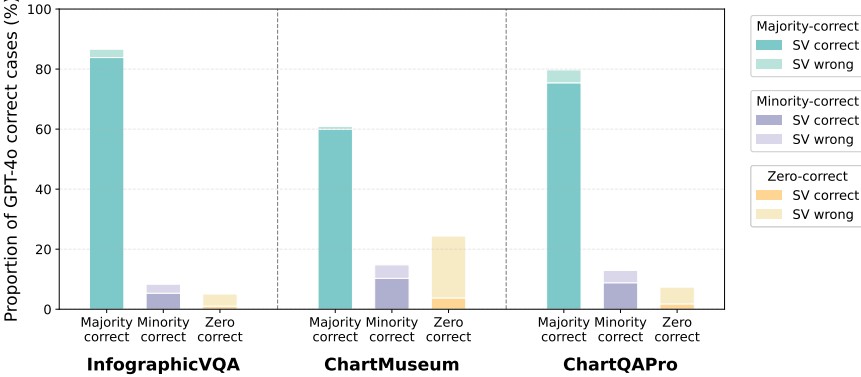

Figure 9: SV's correction ability on verdict's correct cases (GPT-4o as verdict), complementary to its error cases in Figure 4.

Table 11: Results on test sets of InfographicVQA with 7–9B draft experts. We compare SV against closed-source VLMs, open-source VLMs, and tool-driven methods. † denotes results reported in the original papers; all other results are reproduced by us. The best results for each benchmark are highlighted in **bold** and the second-best results are underlined.

| Model | Param Size | InfographicVQA ANLS |
|---|---|---|
| *Closed-source VLMs* | | |
| GPT-4o | – | 76.5 |
| GPT-4o-mini | – | 67.2 |
| *Open-source VLMs* | | |
| Qwen2.5-VL-Instruct | 7B | 79.8 |
| LLaVA-OneVision-1.5 | 8B | 70.3 |
| InternVL3 | 8B | 72.3 |
| Eagle 2.5 | 8B | 74.5 |
| Ovis2.5 | 9B | 81.7 |
| *Tool-driven method* | | |
| DeepEyes | 7B | 75.5 |
| Pixel-Reasoner | 7B | 84.0† |
| **SV w/ GPT-4o Verdict** | – | **86.3** |
| Δ (vs. GPT-4o) | – | +9.8 |

Table 12: Results on test sets on InfographicVQA with 2–4B draft experts. Same evaluation and marking conventions as Table 11.

| Model | Param Size | InfographicVQA ANLS |
|---|---|---|
| *Closed-source VLMs* | | |
| GPT-4o | – | 76.5 |
| GPT-4o-mini | – | 67.2 |
| *Open-source VLMs* | | |
| Qwen2.5-VL-Instruct | 3B | 64.9 |
| LLaVA-OneVision-1.5 | 4B | 67.1 |
| InternVL3.5 | 4B | 74.4 |
| Gemma 3 | 4B | 36.0 |
| Ovis2.5 | 2B | 75.0 |
| *Tool-driven method* | | |
| DeepEyes | 7B | 75.5 |
| Pixel-Reasoner | 7B | 84.0† |
| **SV w/ GPT-4o Verdict** | – | **84.5** |
| Δ (vs. GPT-4o) | – | +8.0 |

# E    ABLATION STUDY ON MODEL POOL COMPOSITIONS

Beyond the fixed model pool used in the main experiments, we further examine SV's generalizability across different model pool compositions by testing on pools with varying model sizes and capabilities. The results show that SV successfully leverages reasoning paths from lightweight models, delivering strong performance while maintaining cost efficiency.

**Evaluation with 7-9B Model Pool (Non-reasoning).** SV maintains its effectiveness when replacing reasoning models with faster non-reasoning alternatives. Specifically, we replace the two reasoning models in our original pool (i.e., GLM-4.1V-9B-Thinking (Team et al., 2025b) and MiMo-VL-7B-RL (Xiaomi, 2025)) with non-reasoning models (i.e., LLaVA-OneVision-1.5-8B (An et al., 2025), and Eagle 2.5-8B (Chen et al., 2025)), while keeping the remaining three models unchanged. While these substitutes sacrifice some reasoning capability, they enable faster inference. As shown in Table 11, with GPT-4o as verdict, SV achieves 86.3% on InfographicVQA under this configuration, surpassing all baselines. Notably, SV surpasses the best draft expert by 4.6% and improves over GPT-4o by 9.8%. These results demonstrate that SV achieves strong performance by integrating reasoning paths from individually weaker but faster models.

**Evaluation with 2-4B Model Pool.** We also evaluate SV on an even smaller model pool consisting of 2-4B models: Qwen2.5-VL-3B-Instruct (Bai et al., 2025), LLaVA-OneVision-1.5-4B (An et al., 2025), InternVL3.5-4B (Wang et al., 2025a), Gemma 3-4B (Team et al., 2025a), and Ovis2.5-2B (Lu et al., 2025). As shown in the Table 12, with GPT-4o as verdict, SV achieves 84.5% on InfographicVQA, surpassing the best draft expert by 9.5% and yielding an 8% gain over GPT-4o. This demonstrates SV's ability to extract effective collective reasoning even from significantly weaker individual models, confirming the robustness of our paradigm across varying model scales.

# F    ANALYSIS OF CONSENSUS SELECTION MECHANISM

This section provides a detailed analysis of our consensus expert selection mechanism, addressing three aspects: (1) selection frequency distribution, (2) impact of normalization on consensus scores, and (3) impact of NLL estimation strategy.

## F.1    SELECTION FREQUENCY DISTRIBUTION

To understand how the consensus mechanism selects draft experts, we analyze the selection frequency of each model across benchmarks. Table 13 reports the proportion of instances in which each model is selected. All models participate with non-trivial frequencies (38.6%–84.7%), with no model dominating or being marginalized. This indicates that different drafts specialize in different subsets of questions, and the consensus mechanism leverages complementary agreement and disagreement across experts rather than collapsing to a single always-selected model.

Table 13: Selection frequency of each draft model across benchmarks.

| Model | InfographicVQA | ChartQAPro |
|---|---|---|
| Qwen2.5-VL-7B-Instruct | 84.7% | 77.7% |
| GLM-4.1V-9B-Thinking | 38.6% | 69.3% |
| MiMo-VL-7B-RL | 54.8% | 56.8% |
| InternVL3-8B | 73.4% | 53.1% |
| Ovis2.5-9B | 48.5% | 43.1% |

## F.2    IMPACT OF NORMALIZATION ON CONSENSUS SCORES

We further ablate the effect of normalization on consensus scores, which is motivated by the calibration gap across draft experts mentioned in Section 3.4. Different VLMs produce perplexity scores on systematically different numerical scales due to training and tokenization differences. In our pool, for example, Qwen2.5-VL and GLM-4.1V-Thinking tend to output larger-magnitude NLLs. Without normalization, such scale mismatch causes high-magnitude models to dominate the consensus even when they do not genuinely agree more often, thereby reducing selection diversity.

We evaluate normalization on two draft pools on InfographicVQA: (i) the main pool used in our primary experiments, mixing reasoning and non-reasoning drafts, and (ii) an additional pool consisting only of non-reasoning drafts, introduced in Appendix E. Tables 14a and 14c show that, on the main pool, removing normalization collapses selection to a near-fixed subset: Qwen and GLM are selected in almost all instances. This collapse hurts performance on ChartQAPro, demonstrating that calibration artifacts can actively degrade consensus quality. Although InfographicVQA remains similar without normalization, this results from an accidental strong pairing being repeatedly selected rather than a principled aggregation.

To verify that this trend is not pool-specific, we repeat the ablation on the additional non-reasoning pool. Tables 14b and 14d show the same selection-collapse pattern without normalization, together with a clear performance drop. Across both pools, normalization mitigates calibration bias, prevents dominance by a few drafts, and yields more stable performance.

Table 14: Normalization ablations on the main and additional draft pools.

(a) Ablations on normalization on the main draft pool.

| Variant | InfographicVQA | ChartQAPro |
|---|---|---|
| SV w/o norm. | 88.9 | 59.4 |
| SV | 88.4 | 64.0 |

(b) Ablations on normalization on the additional non-reasoning pool.

| Variant | InfographicVQA |
|---|---|
| SV w/o norm. | 84.6 |
| SV | 86.3 |

(c) Selection frequency on the main pool without normalization.

| Model | InfographicVQA | ChartQAPro |
|---|---|---|
| Qwen2.5-VL-7B-Instruct | 99.9% | 100.0% |
| GLM-4.1V-9B-Thinking | 77.1% | 99.9% |
| MiMo-VL-7B-RL | 57.0% | 71.5% |
| InternVL3-8B | 47.0% | 21.4% |
| Ovis2.5-9B | 19.1% | 2.2% |

(d) Selection frequency on the additional non-reasoning pool.

| Model | w/ norm. | w/o norm. |
|---|---|---|
| Qwen2.5-VL-7B-Instruct | 87.3% | 99.9% |
| Eagle2.5-8B | 65.4% | 98.0% |
| LLaVA-OneVision-1.5-8B | 28.6% | 79.3% |
| InternVL3-8B | 74.5% | 16.7% |
| Ovis2.5-9B | 44.2% | 6.11% |

### F.3 IMPACT OF NLL ESTIMATION STRATEGY

We examine whether computing NLL only on final answers introduces off-policy bias, since draft experts' answers are generated together with reasoning trajectories. Table 15 compares expert selection using answer-only NLL (SV) versus full-trajectory NLL. The two variants achieve very similar performance, suggesting that any off-policy bias from answer-only scoring is negligible for this task. Moreover, answer-only scoring is more computationally efficient, as it avoids computing NLL over long reasoning traces with many extra tokens; answer-only NLL also provides a cleaner signal by avoiding noise from diverse reasoning styles across models. We therefore use answer-only NLL in the main experiments.

Table 15: Ablation on NLL estimation strategy.

| Scoring variant | InfographicVQA | ChartQAPro |
|---|---|---|
| Answer+reasoning NLL | 87.9 | 64.3 |
| Answer-only NLL (SV) | 88.4 | 64.0 |

## G COMPARISON TO LLAVA-CRITIC (LMM-AS-A-JUDGE BASELINE)

We compare SV to a learned LMM-as-a-Judge baseline, LLaVA-Critic (Xiong et al., 2025). LLaVA-Critic is trained as a generalist multimodal evaluator that jointly leverages visual evidence and textual reasoning to score or rank multiple candidates.

**Experimental setup.** We follow the official LLaVA-Critic evaluation template and consider two judging modes: (i) pointwise scoring, where the judge assigns a scalar score to each candidate and selects the highest-scored one; (ii) pairwise ranking, where candidates are compared (i.e., first two candidates are compared, and the winner is then compared against the third in our scenario). To ensure a comprehensive comparison, we evaluate LLaVA-Critic under two settings: (1) judging the same three consensus experts selected by SV (directly replacing the verdict); (2) judging five candidate models. In all cases, the judge outputs a single best candidate.

For SV, we conduct experiments with two verdict models: GPT-4o and LLaVA-OneVision-7B (Li et al., 2024). GPT-4o serves as a strong proprietary verdict model, matching our main experiments, while LLaVA-OneVision-7B shares the same backbone as LLaVA-Critic, enabling a fairer comparison in the open-source regime.

**Results.** Table 16 shows that when judging the same three draft experts, SV consistently outperforms LLaVA-Critic-7B across all three benchmarks under both verdict choices. With GPT-4o as the verdict, SV improves over LLaVA-Critic-7B by 4.9–11.9%. With LLaVA-OneVision-7B as the verdict, SV still yields gains of 0.5–6.6%, which demonstrates that the advantage stems from SV's design rather than from verdict model superiority alone.

Table 16: Performance of SV and LLaVA-Critic-7B.

| Method | InfographicVQA | ChartMuseum | ChartQAPro |
|---|---|---|---|
| LLaVA-Critic-7B (pointwise) | 83.5 | 40.1 | 52.4 |
| LLaVA-Critic-7B (pairwise) | 81.4 | 38.9 | 52.1 |
| SV w/ GPT-4o verdict | **88.4** | **49.3** | **64.0** |
| SV w/ LLaVA-OneVision-7B verdict | 84.0 | 44.1 | 59.0 |

Table 17: Average judge/verdict tokens per sample.

| Method | InfographicVQA | ChartMuseum | ChartQAPro |
|---|---|---|---|
| LLaVA-Critic-7B (pointwise) | 1053.3 | 3723.7 | 1290.7 |
| LLaVA-Critic-7B (pairwise) | 1342.9 | 4689.4 | 1759.1 |
| SV w/ GPT-4o verdict | 701.7 | 3122.3 | 1441.7 |
| SV w/ LLaVA-OneVision-7B verdict | 702.4 | 3123.0 | 1442.4 |

Table 18: Performance of SV and LLaVA-Critic-7B judging five candidate models (1k samples).

| Method | InfographicVQA | ChartMuseum | ChartQAPro |
|---|---|---|---|
| LLaVA-Critic-7B (pointwise) | 82.3 | 34.6 | 56.5 |
| SV w/ GPT-4o verdict | **88.3** | **49.3** | **63.4** |
| SV w/ LLaVA-OneVision-7B verdict | 83.6 | 44.1 | 57.8 |

Table 19: Performance of SV and LLaVA-Critic-72B judging three consensus experts (1k samples).

| Method | InfographicVQA | ChartMuseum | ChartQAPro |
|---|---|---|---|
| LLaVA-Critic-72B (pointwise) | 82.6 | 40.4 | 53.5 |
| LLaVA-Critic-72B (pairwise) | 83.1 | 43.0 | 55.8 |
| SV w/ GPT-4o verdict | **88.3** | **49.3** | **63.4** |
| SV w/ LLaVA-OneVision-7B verdict | 83.6 | 44.1 | 57.8 |

This gap is attributed to the different aggregation principles: LLaVA-Critic performs *selection*-based judging and can only pick one trajectory, while SV performs *synthesis*-based verification, cross-checking and integrating complementary factual cues across trajectories. In practice, the critic is also more sensitive to surface form (e.g., verbosity, repetitiveness, or stylistic fluency), whereas SV is guided by consensus evidence and is therefore more robust to such artifacts.

Table 17 further reports the total judge/verdict tokens. Under comparable prefill-based budgets, SV achieves higher performance with similar or lower token usage on InfographicVQA and ChartMuseum, and still attains a much better cost–performance trade-off on ChartQAPro.

When increasing the candidate pool to five models (Table 18), LLaVA-Critic-7B remains clearly below SV, indicating that simply adding more candidates to a selection-based judge does not close the gap. Finally, even a much larger judge (LLaVA-Critic-72B, Table 19) is still notably weaker than SV, even including the variant with the smaller LLaVA-OneVision-7B verdict. This confirms that SV is more effective than selection-based judging for information-intensive visual reasoning.

## H ABLATION STUDIES ON VERDICT INPUT CONFIGURATION

### H.1 IMPACT OF VISUAL INPUT TO VERDICT

We examine whether visual input is necessary for the verdict or if reasoning paths alone suffice. Table 20 presents results where the verdict receives only textual reasoning paths without image input. The results show that SV without visual input achieves modest gains over the reasoning baseline on InfographicVQA, and even underperforms on ChartMuseum and ChartQAPro. In contrast, incorporating visual input for verdict yields substantial improvements across all benchmarks. These results demonstrate that visual grounding is essential for the verdict to cross-check the factual accuracy of extracted information and distinguish correct from incorrect interpretations of the image.

Table 20: Ablations on visual input to the verdict GPT-4o.

| Method | InfographicVQA ANLS | ChartMuseum Acc | ChartQAPro Acc |
|---|---|---|---|
| Reasoning baseline | 84.8 | 48.0 | 57.3 |
| GPT-4o Verdict w/o input | 85.9 | 47.1 | 53.2 |
| GPT-4o Verdict w/ input (SV) | **88.4** | **49.3** | **64.0** |

Table 21: Ablations on additional structured image input to the verdict GPT-4o.

| Method | InfographicVQA ANLS | ChartMuseum Acc | ChartQAPro Acc |
|---|---|---|---|
| Reasoning baseline | 84.8 | 48.0 | 57.3 |
| GPT-4o Verdict w/o input | 88.3 | **49.5** | 59.4 |
| GPT-4o Verdict w input (SV) | **88.4** | 49.3 | **64.0** |

## H.2 IMPACT OF STRUCTURED IMAGE INPUT TO VERDICT

In our experimental setup in Section 4.1, we preprocess each image via PP-StructureV3, a document parsing model that generates Markdown representations capturing layout, textual blocks, and visual metadata (Cui et al., 2025). This structured representation is then rendered as an image and provided as an additional image input for the verdict. This allows the verdict to access both the raw visual content and a layout-aware text representation simultaneously. To verify whether this input is critical or merely auxiliary, we conduct an ablation study (Table 21).

The results show that SV achieves substantial gains over the reasoning baseline even without structured input. With the structured input, performance is generally slightly improved, though the gain is negligible or even marginally lower in some cases. This pattern suggests that structured OCR-derived signals are not essential for SV's core performance, but may assist the verdict to distinguish among competing reasoning paths.

## I QUANTITATIVE ANALYSIS OF DRAFT COMPLEMENTARITY

To characterize the complementarity of draft experts, we focus on minority-correct recovery cases on InfographicVQA where only one of the three selected experts is correct and SV subsequently recovers the correct answer. These cases are most diagnostic for understanding how specific models provide unique, correct information and how others behave to help the verdict distinguish cues.

We randomly sample 50 minority-correct recovery instances and manually categorize their dominant reasoning bottlenecks into five types, summarized in Table 22. Extraction-related failures account for half of the cases, followed by color matching and global scan.

Table 22: Distribution of dominant reasoning bottlenecks over 50 minority-correct recovery cases on InfographicVQA.

| Reasoning bottleneck | Description | Frequency (%) |
|---|---|---|
| Extraction | Locating and reading fine-grained text/numbers | 50 |
| Color matching | Matching colors in legends/charts to labels | 18 |
| Global scan | Aggregating evidence across the full image | 16 |
| Localization | Finding query-relevant regions | 10 |
| Numerical comparison | Comparing numerical values | 4 |

Table 23 reports per-model success rates within each bottleneck category over the 50 minority-correct instances. We observe a clear division of labor across experts: GLM-4.1V-Thinking and MiMo-VL-RL are most reliable for fine-grained extraction (58% and 70% success, respectively), with MiMo additionally strong on global-scan cases (75%). This advantage is consistent with their reasoning-oriented behavior: their step-by-step trajectories enable iterative verification of extracted values and cross-checking across multiple regions. Ovis2.5 and GLM-4.1V-Thinking perform best on color matching (57% and 60%), while Qwen2.5-VL dominates localization (80%) and numer-

ical comparison (100%). We find that other experts often fail in these categories due to keyword-comprehension errors, whereas Qwen2.5-VL's correct grounding makes its answer salient for the verdict to identify even when other models provide detailed but misdirected reasoning. Overall, no single expert dominates all reasoning types, confirming that SV benefits from specialized and complementary strengths rather than redundant voting.

Table 23: Per-model success rates (%) on minority-correct recovery cases, broken down by reasoning bottleneck.

| Reasoning type | Qwen2.5-VL-Instruct | GLM-4.1V-Thinking | MiMo-VL-RL | Ovis2.5 | InternVL3 |
|---|---|---|---|---|---|
| Extraction | 15 | 58 | **70** | 38 | 15 |
| Color matching | 0 | **60** | 33 | 57 | 25 |
| Localization | **80** | 0 | 0 | 33 | 0 |
| Global scan | 0 | 33 | **75** | 20 | 0 |
| Numerical comparison | **100** | 0 | 0 | 0 | 0 |

Finally, we summarize the most frequent expert combinations all 243 minority-correct recovery cases on InfographicVQA, as shown in Table 24. Two typical complementarity patterns emerge. First, balanced visual-skill combinations (e.g., Qwen–Ovis–InternVL) appear in cases requiring diverse perceptual cues. Second, extraction-focused pairings (e.g., Qwen–MiMo–InternVL and Qwen–GLM–Ovis) arise in extraction-intensive problems, where reasoning drafts provide accurate fine-grained values and non-reasoning drafts contribute robust localization for cross-verification. Overall, this analysis supports our claim that SV does not simply average similar models: it exploits complementary reasoning strengths across draft experts, and the minority-correct recoveries we observe are closely associated with diverse and specialized reasoning trajectories across models.

Table 24: Most frequent expert combinations among all minority-correct cases on InfographicVQA.

| Rank | Model combination | Frequency (%) |
|---|---|---|
| 1 | Qwen + Ovis + InternVL | 33.3 |
| 2 | Qwen + MiMo + InternVL | 14.0 |
| 3 | Qwen + GLM + Ovis | 11.1 |

## J  FAILURE MODE ANALYSIS OF SV

We analyze cases where the majority of selected draft experts are correct but SV produces an incorrect answer on InfographicVQA with GPT-4o as verdict. These instances isolate failures when the consensus mechanism or verdict synthesis fails despite having correct information available. Under exact-match scoring, such cases account for 3.98% of majority-correct examples; after normalizing answer formats (e.g., "Feb 11" vs. "February 11"), the true failure rate is 1.34% (34 cases).

**Failure modes.** Two dominant patterns emerge. (i) Fine-grained extraction errors (52.9%) arise from tiny numbers or small text on very tall infographics. Although draft experts often extract the correct cue, GPT-4o may fail to verify it because its tile-based high-resolution pipeline downsamples long inputs, losing subtle details. (ii) Color matching errors (35.3%) require aligning colored legends or regions with labels. This reflects a shared VLM capability gap: when both drafts and verdict struggle with color discrimination, drafts provide uncertain reasoning (sometimes explicit guesses), and GPT-4o exhibits verbosity bias in 58.3% of such cases (i.e., it follows the draft with the longest trajectory despite majority consensus pointing elsewhere).

Overall, SV inherits current VLM weaknesses in precise color reasoning and fine-grained perception, and when visual evidence is ambiguous the verdict may over-weight fluent yet incorrect drafts.

## K  FAILURE MODE ANALYSIS OF TOOL-DRIVEN PIPELINE

As mentioned in Section 2, tool-driven methods represent a line of work that augments vision-language reasoning with explicit zoom-in operations. The representative pipeline DeepEyes is designed to iteratively ground into image regions, and integrate them into the ongoing reasoning trajec-

Table 25: Breakdown of SV failure modes on InfographicVQA.

| Failure mode | Proportion |
|---|---|
| Fine-grained extraction | 52.9% |
| Color matching | 35.3% |
| Other | 11.8% |

tory under an RL framework. This mechanism has proven effective on high-resolution benchmarks, where localized inspection of fine details is crucial.

However, DeepEyes is not specifically trained on our benchmarks, which require reasoning over information-intensive images with densely interleaved textual and visual elements. Its performance on InfographicVQA reveals the current limitations of such tool-based pipelines in this domain. We categorize the observed deficiencies into three core challenges:

(i) Tendency toward literal grounding. DeepEyes is proficient at small-scale grounding but often focuses on literal text spans or legends rather than reasoning-critical regions. For example, when a question requires aligning numerical values with a chart axis, the model frequently grounds directly onto the answer text or nearby labels instead of the relevant data regions. This shortcut strategy works for simple queries but fails on complex reasoning on information-intensive images that require global comparison.

(ii) Inefficient tool usage. Although DeepEyes is trained to iteratively apply zoom-in tools, we observe that it invokes only one zoom step in more than half of the test cases. Among the double-zoom cases, 92.8% duplicate the same bounding box, which serves only for verification rather than exploration. In some instances, the model zooms into empty areas or irrelevant regions.

(iii) Lack of robustness on long and dense images. Information-intensive images often contain multi-panel figures and dense annotations. DeepEyes cannot maintain a trajectory across multiple zoom steps, making it difficult to integrate dispersed evidence. As a result, tasks requiring cross-region synthesis, such as counting, sorting, or comparing across multiple subplots, remain challenging.

Overall, this analysis indicates that while tool-driven pipelines are promising for high-resolution inspection tasks, they face notable difficulties applying to information-intensive images without domain-specific supervision. In contrast, SV achieves strong performance without additional training, offering a simple and effective alternative for reasoning over complex multimodal inputs.

## L    QUALITATIVE EXAMPLE

Figure 10 illustrates a case where all three draft experts produced incorrect reasoning paths, yet the verdict successfully corrected the answer. Specifically, the draft experts faced different types of failures: some mis-extracted information from the image, others extracted the key information correctly but failed to sort the values properly, and thus all generated wrong answers. Interestingly, the verdict itself, when asked directly, also tends to answer "Australia" incorrectly. However, when analyzing the noisy and conflicting reasoning paths together, the verdict was able to recover the correct answer (Portugal).

This example complements the main results section: while Figure 3 illustrates recovery from minority-correct experts, here we present a zero-correct case to show that SV can still synthesize the correct solution even when all drafts and the verdict individually fail.

## M    PROMPT TEMPLATES

### M.1    CHAIN-OF-THOUGHT PROMPTS

As described in Section 4.1, we employ a Chain-of-Thought prompt for each consensus expert to generate reasoning paths and apply it identically when evaluating baselines. For InfographicVQA and HR-Bench 4K, we use the same CoT prompt. For ChartMuseum (Tang et al., 2025), we adopt its official reasoning prompt, and adapt that prompt strategy to ChartQAPro, given their similarity in task complexity. Since ChartQAPro requires different prompt templates tailored to question

types (Masry et al., 2025), we first follow its official template per question type, then concatenate it with our reasoning prompt.

The reasoning prompts for these datasets are shown in Figure 11.

### M.2 PROMPTS FOR VERDICT

The user prompts used in the verdict stage are identical across datasets except for the final instruction sentence, which is customized (see Figure 13). For GPT-4o as verdict, the system prompt is shown in Figure 12. For Qwen-2.5-VL-72B-Instruct as verdict, we prepend its system prompt at the beginning of the user prompt.

## N  THE USE OF LARGE LANGUAGE MODELS (LLMs)

In this work, we used LLMs solely for auxiliary tasks such as language polishing, prompt refining, and proofreading. Importantly, these interventions did not contribute any main scientific insight, experimental design, or methodological advance. All core ideas, experiments, analyses, and claims in this paper are the work of the authors.

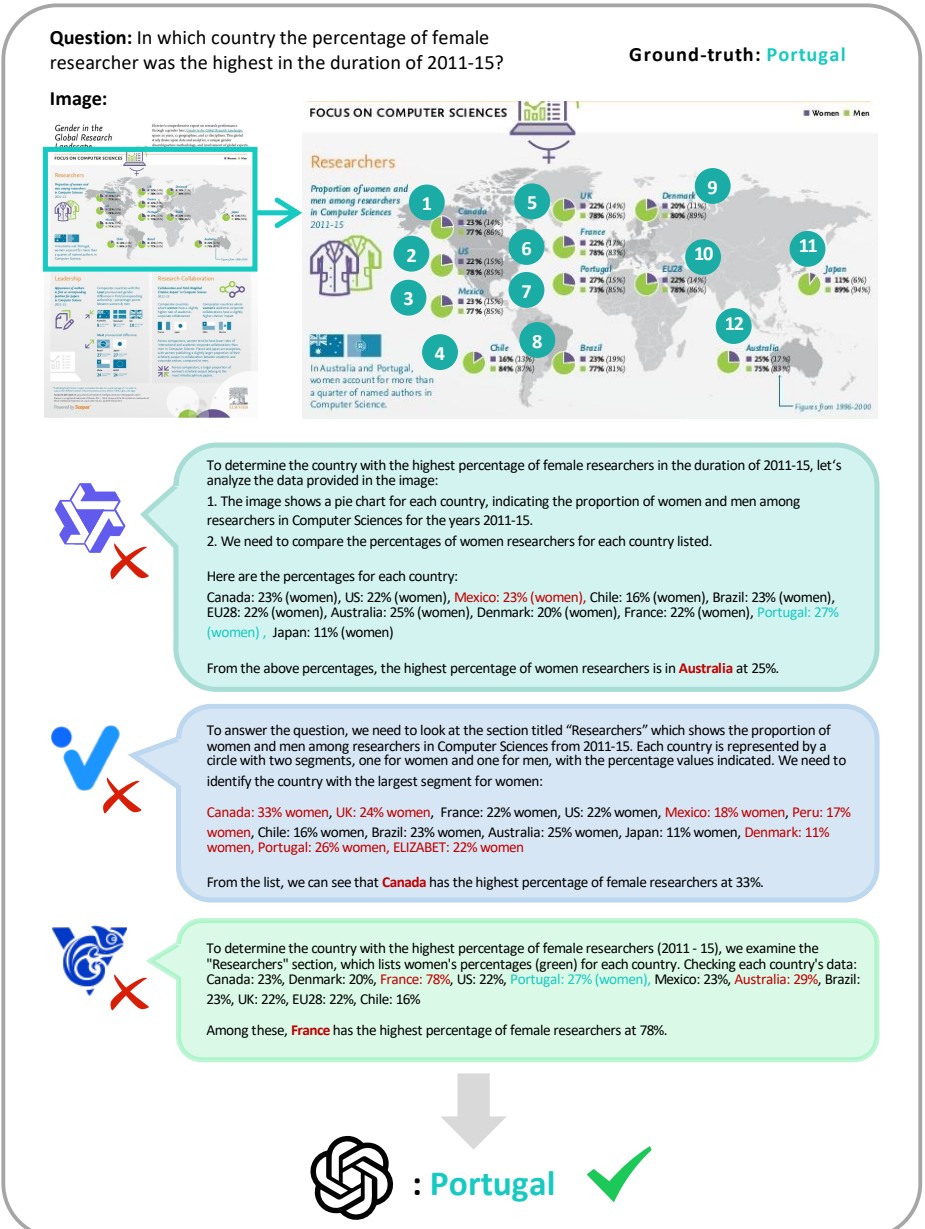

Figure 10: A qualitative zero-correct case corrected by verdict. All three draft experts fail due to errors in extracting or sorting visual information, yet the verdict synthesizes their noisy reasoning paths to recover the correct answer (i.e., Portugal).

> **InfographicVQA / HR-Bench 4K**
>
> ```
> Question: {QUESTION} Please think step-by-step about the image to
> answer the question using a single word or phrase enclosed within \\
> boxed{{}}.
> ```

> **ChartMuseum**
>
> ```
> Please answer the question using the chart image.
>
> Question: {QUESTION}
>
> Please first generate your reasoning process and then provide the
> user with the answer. Use the following format:
>
> <think>
> ... your thinking process here ...
> </think>
> <answer>
> ... your final answer (entity(s) or number) ...
> </answer>
> ```

> **ChartQAPro**
>
> ```
> {PROMPT for a specific question type}
>
> Please first generate your reasoning process and then provide the
> user with the answer. Use the following format:
>
> <think>
> ... your thinking process here ...
> </think>
> <answer>
> ... your final answer (entity(s) or number) ...
> </answer>
> ```

Figure 11: Prompt templates for reasoning.

> **All benchmarks**
>
> ```
> You are a vision-and-language judge. Follow the instructions strictly
> .
> ```

Figure 12: System prompt template for verdict.

InfographicVQA / ChartMuseum

```
Question:
{QUESTION}
--- Model 1 ---
Reasoning:
{Reasoning path 1}
Proposed Answer: {Answer 1}
--- Model 2 ---
Reasoning:
{Reasoning path 2}
Proposed Answer: {Answer 2}
--- Model 3 ---
Reasoning:
{Reasoning path 3}
Proposed Answer: {Answer 3}
Given the raw image, the layout-annotated image, the question, and
the reasoning from three models, please give the final answer using a
 single word or phrase enclosed within \\boxed{{}}.
```

ChartQAPro

```
Question:
{QUESTION}
--- Model 1 ---
Reasoning:
{Reasoning path 1}
Proposed Answer: {Answer 1}
--- Model 2 ---
Reasoning:
{Reasoning path 2}
Proposed Answer: {Answer 2}
--- Model 3 ---
Reasoning:
{Reasoning path 3}
Proposed Answer: {Answer 3}
Given the raw image, the layout-annotated image, the question, and
the reasoning from three models, please directly give the final
answer enclosed within \\boxed{{}}.
```

HR-Bench 4K

```
Question:
{QUESTION}
--- Model 1 ---
Reasoning:
{Reasoning path 1}
Proposed Answer: {Answer 1}
--- Model 2 ---
Reasoning:
{Reasoning path 2}
Proposed Answer: {Answer 2}
--- Model 3 ---
Reasoning:
{Reasoning path 3}
Proposed Answer: {Answer 3}
Given the image, the question, and the reasoning from three models,
please directly give the final answer with the option's letter
enclosed within \\boxed{{}}.
```

Figure 13: User prompt templates for verdict.

