# OpenReview forum: "Small Drafts, Big Verdict: Information-Intensive Visual Reasoning via Speculation"
_ICLR.cc/2026/Conference — ICLR 2026 Poster_

### Official Review · Reviewer_E8fc · 2025-10-29

**Soundness:** 3
**Presentation:** 3
**Contribution:** 2
**Rating:** 4
**Confidence:** 4

**Summary:**

This paper introduces SV, a training-free framework that combines multiple lightweight draft experts with a large verdict model. They use small VLMs to generate reasoning paths as candidates, and use strong VLMs to produce the final answer. They claim SV achieves consistent gains over many benchmarks.

**Strengths:**

1. Enhancing visual reasoning ability is a highly important and actively studied research problem.

2. The paper is easy to follow, the figures are clear, and the code is open-sourced.

3. The experiments are detailed, and the ablation study in Section 4.4 is very comprehensive.

**Weaknesses:**

1. My main concern with this work lies in the evaluation. If this issue can be properly addressed, I would be willing to raise my score. Currently, Table 1 compares the performance of several VLMs, and SV claims to outperform all of them. However, I would like to see a cost analysis. Although Appendix B provides some numerical results, it lacks a comparison of the computational cost between SV and the baseline methods. How much additional cost does SV introduce compared to the baselines? Under the same computational cost, would using only the verdict yield a similar effect? Likewise, under the same cost, would conducting debates among multiple draft models achieve comparable results?

2. The literature review is missing several important works. In the area of vision-language model reasoning, this paper is not limited to tool-related methods; therefore, it should also include more general VLM reasoning studies and cite some representative works in that domain.

**Questions:**

1. Address the issues mentioned in the weaknesses.

2. The formatting on page 17 could be improved.

---

> ### Author Response · Authors · 2025-11-21
> **Response to Reviewer E8fc (Part 1/2)**
>
> We thank you for the valuable feedback! We have revised the paper accordingly and addressed each of the raised weaknesses and questions in detail below. All revisions are marked in blue in the revised manuscript.
>
> ### **W1 & Q1: Computational cost analysis of SV vs. baselines**
>
> We sincerely appreciate this important concern and agree that cost-efficiency analysis is crucial for practical deployment. Following your suggestion, we have conducted a detailed cost analysis (now added in Section 4.6 and Appendix C.2). In summary, experiments show SV achieves superior cost-efficiency compared with both strong verdict models and draft-model debate baselines.
>
> 1. **SV vs. verdict-only baselines:**
>
>     To compare SV with calling only the verdict model, we compare the total API cost for SV, GPT-4o, and o1. Because o1 is very expensive, we randomly sample 1,000 instances from each dataset. The results in Tables 1 and 2 show that:
>
>     - Compared with 4o, SV achieves **6.6-12.0% improvement over GPT-4o while maintaining comparable cost**.
>     - Compared with o1, SV substantially outperforms on InfographicVQA and ChartQAPro, and is comparable on ChartMuseum, **while using only 15–26% of o1’s cost.**
>
>     Table 1: Performance comparison on 1000 random test samples.
>
>     | Method | InfographicVQA | ChartMuseum | ChartQAPro |
>     | --- | --- | --- | --- |
>     | GPT-4o | 76.3 | 42.7 | 51.7 |
>     | o1 | 77.8 | **50.6** | 58.8 |
>     | **SV w/ GPT-4o Verdict** | **88.3** | 49.3 | **63.4** |
>
>     Table 2: Average inference API cost per sample across benchmarks.
>
>     | Method | InfographicVQA | ChartMuseum | ChartQAPro |
>     | --- | --- | --- | --- |
>     | GPT-4o | $0.0038 | $0.0213 | $0.0210 |
>     | o1 | $0.0263  | $0.0663 | $0.0478 |
>     | **SV w/ GPT-4o Verdict** | $0.0068 | $0.0109 | $0.0071 |
> 2. **SV vs. Multi-Draft Debate:**
>
>     For the debate baseline with a similar cost, we implemented a “draft-model debate” setup where we kept the draft stage identical to SV and only changed the verdict model to a draft model (i.e, GLM-4.1V-9B-Thinking, MiMO-VL-7B-RL, or Qwen2.5-VL-7B-Instruct). As the only difference in the token costs is the verdict output tokens, we prompt the verdict model to output only the final answer, as in our main experiments, to keep token counts comparable.
>
>     The results in Tables 3 and 4 show that:
>
>     - **Using draft models as verdicts (debate baselines) consistently underperforms SV across all datasets.**
>     - The reasoning draft models (i.e., GLM-4.1V-9B-Thinking and MiMO-VL-7B-RL) **produce 60-200x more output tokens, yet still underperform SV.**
>     - Under similar token costs, Qwen2.5-VL-7B-Instruct delivers evident weaker performance than SV.
>
>     Table 3: Performance with different verdict models.
>
>     | Verdict | InfographicVQA | ChartMuseum | ChartQAPro |
>     | --- | --- | --- | --- |
>     | GLM-4.1V-9B-Thinking | 84.7 | 48.0 | 59.4 |
>     | MiMO-VL-7B-RL | 85.4 | 46.9  | 60.3 |
>     | Qwen2.5-VL-7B-Instruct | 84.1  | 47.1 | 57.2 |
>     | **GPT-4o (SV)** | **88.4** | **49.3** | **64.0** |
>
>     Table 4: Average verdict output tokens per sample across benchmarks.
>
>     | Verdict | InfographicVQA | ChartMuseum | ChartQAPro |
>     | --- | --- | --- | --- |
>     | GLM-4.1V-9B-Thinking | 272.1 | 604.5 | 440.5 |
>     | MiMO-VL-7B-RL | 183.2 | 368.2 | 378.1 |
>     | Qwen2.5-VL-7B-Instruct | 3.3 | 3.4 | 2.8 |
>     | **GPT-4o (SV)** | 2.7 | 3.0 | 2.2 |
>
> Our analysis shows that SV achieves great cost-efficiency: it reaches or surpasses GPT-o1-level performance while using only 15–26% of its cost, and it significantly outperforms GPT-4o and multi-draft debates under comparable or lower computational budgets.

---

> ### Author Response · Authors · 2025-11-21
> **Response to Reviewer E8fc (Part 2/2)**
>
> ### **W2 & Q1: Related work on general VLM reasoning**
>
> We thank the reviewer for pointing out the missing discussion of general VLM reasoning beyond tool-based approaches. In the revised version,  we have expanded Section 2 by adding a new paragraph **“General Vision–Language Model Reasoning”** that covers discussion on:
>
> - **Prompt-enhanced methods** that use chain-of-thought prompting for structured reasoning;
> - **RL-enhanced methods** that employ supervised fine-tuning and reinforcement learning to optimize multi-step reasoning trajectories;
> - **Agentic frameworks** that treat VLMs as planners that decompose queries and interactively choose actions via explicit visual tools or implicit latent-space reasoning.
>
> We also highlight that some methods rely on potentially imprecise or weakly supervised visual operations, making them vulnerable to error propagation; SV offers an alternative by cross-validating draft experts under a verdict to filter out erroneous information and improve robustness in such cases.
>
>
> ### **Q2: Formatting on page 17**
>
> Thank you for the feedback! In the revised manuscript (now on page 26), we have adjusted the layout to remove the excessive whitespace and place the two figures consecutively.

---

> > ### Comment · Reviewer_E8fc · 2025-11-21
> >
> > Thanks for the timely rebuttal. I think the cost analysis and the newly added related work have addressed my concerns. I am happy to raise my score to 6.

---

> > > ### Author Response · Authors · 2025-11-22
> > >
> > > Thank you for the positive feedback and for raising the score!

---

### Official Review · Reviewer_656R · 2025-10-31

**Soundness:** 3
**Presentation:** 3
**Contribution:** 3
**Rating:** 6
**Confidence:** 4

**Summary:**

This paper addresses the difficulty of VLMs in reasoning over information-intensive images that combine dense text and fine-grained graphics (e.g., infographics, charts), and introduces Speculative Verdict (SV), a training-free framework for information-intensive visual reasoning tasks. Inspired by speculative decoding, SV operates in two stages. The first is a draft stage where multiple lightweight VLMs generate diverse reasoning paths; and the second is a verdict stage where a large VLM synthesizes these paths to produce the final answer. The authors introduce a consensus expert selection mechanism that forwards only high-agreement reasoning paths to the verdict model. SV is evaluated on several benchmarks (InfographicVQA, ChartMuseum, ChartQAPro, and HR-Bench 4K) and demonstrates consistent improvements over strong baselines while maintaining cost efficiency.

**Strengths:**

- Originality: The paper presents a novel adaptation of speculative decoding for visual reasoning quality improvement rather than its original purpose of inference acceleration.

- Quality:
  1. The proposed approach is effective.
  2. The experimental evaluation is comprehensive, covering multiple benchmarks and comparing against strong baselines.
  3. The ablation studies provide insights into the importance of different components.

- Clarity:
  1. The paper is well-structured and clearly written.
  2. The figures effectively illustrate key concepts and results. (BTW, I like the animal icons in Figure 2.)
  3. The methodology is described in sufficient detail.

- Significance: The paper addresses a significant challenge in multimodal AI with a cost-effective solution that outperforms more expensive alternatives, which has practical implications for deploying such systems.

**Weaknesses:**

1. The tables lack clarity: The tables lack direct comparison between baseline and SV, such as directly providing the increment from GPT-4o (line 332) to SV+4o (line 341), and also the increment on Qwen2.5VL-72B, so that readers can easily compare the performance gain regarding different base models, instead of refering to other places in the paper (such as refering to line 377). Althought I like the figures and plots, the tables really need improving.

2. Limited analysis of computational efficiency: This paper claimis that SV is more cost-efficient, but doesn't provide detailed metrics comparing computational costs (e.g., FLOPs, inference time, memory usage), and the budget cost.

3. Restricted Draft Model Pool: The evaluation is restricted to a fixed draft model pool, limiting understanding of how SV would perform with a more diverse set of draft models.

4. Insufficient Analysis of Failure Cases: The paper would benefit from a more detailed analysis of cases where SV fails to understand the approach's limitations.

5. Limited Comparison to Advanced Ensemble Methods: The paper compares SV to majority voting but doesn't compare to more advanced ensemble methods that could be applied to this problem.

6. Potential Overfitting to Specific Benchmarks: While evaluating on multiple benchmarks, they all focus on similar types of information-intensive visual reasoning, making generalizability unclear.

**Questions:**

1. Why SV improves more dramatically with GPT-4o (11.9/6.6/11.4/4) compared to Qwen 72B (2.5/7.5/2.3/2.5), especially for InfographicVQA and ChartQAPro?

2. Could you provide more detailed metrics on the computational efficiency of SV compared to baselines? For example, inference time, FLOPs, or memory usage would help quantify the efficiency claims.

3. How sensitive is SV to the choice of draft models? Have you experimented with draft models of different sizes and architectures beyond the ones reported?

4. In what types of cases does SV fail? A more detailed analysis of failure modes would help understand the limitations.

5. How does SV compare to more advanced ensemble methods beyond majority voting? For example, methods that learn to weight or combine the outputs of multiple models.

6.How well does SV generalize to other types of visual reasoning tasks beyond the information-intensive benchmarks evaluated?

---

> ### Author Response · Authors · 2025-11-21
> **Response to Reviewer 656R (Part 1/6)**
>
> We thank the reviewer for the insightful and positive evaluation of our work! We address each of the raised weaknesses and questions below, and all revisions are marked in blue in the revised manuscript.
>
> ### **W1: Table clarity**
>
> We thank the reviewer for the helpful suggestion! In the revised version, we have restructured the main results tables so that the performance gains of SV over its base models (e.g., GPT-4o, Qwen2.5-VL-72B-Instruct) are shown explicitly.
>
> ### **Q1: Why does SV improve more dramatically with GPT-4o compared to Qwen2.5-VL-72B-Instruct?**
>
> We appreciate this insightful observation. The improvement magnitude depends on **how much headroom the verdict model has on a given benchmark** and **how complementary the drafts and verdict are to each other's weaknesses.** SV yields the largest gains when drafts supply missing fine-grained evidence and the verdict is strong enough to verify and synthesize that evidence into a correct answer.
>
> 1. **GPT-4o as verdict:**
>
>     On information-intensive benchmarks, GPT-4o starts from a lower baseline primarily due to a perceptual bottleneck on dense, high-resolution layouts. Its tile-based processing downsamples such inputs, so many errors arise from missed fine-grained cues rather than weak reasoning. SV directly addresses this gap: drafts provide diverse localized extractions and partial cues, which are passed to the verdict as contextual evidence. Given stronger multi-hop verification and reasoning ability, GPT-4o can then cross-check conflicts, consolidate partial evidence, and correct errors. **This is reflected in its stronger recovery on hard cases: GPT-4o recovers +19.6% and +6.2% more minority-correct cases than Qwen2.5-VL-72B-Instruct on InfographicVQA and ChartQAPro** (Tables 1 and 2), which translates into larger baseline gains.
>
> 2. **Qwen2.5-VL-72B-Instruct as verdict:**
>
>     Qwen2.5-VL-72B-Instruct is already specialized for information-rich visual inputs, giving it strong OCR/chart/document localization. So, its baseline on InfographicVQA/ChartQAPro is much higher, and the draft models' similar perception strengths offer limited complementarity, leaving less room for SV to recover perception-driven mistakes. Moreover, when reasoning paths are contradictory or minority-correct, Qwen2.5-VL-72B-Instruct struggles to filter noisy reasoning and identify reliable signals, resulting in a clearly lower recovery advantage than GPT-4o.
>
>
> Table 1: Verdict correction ability on InfographicVQA.
>
> | Model | Minority-correct | Zero-correct |
> | --- | --- | --- |
> | Qwen2.5-VL-72B-Instruct | 38.4% | 2.2% |
> | GPT-4o | 58.0% | 8.9% |
>
> Table 2: Verdict correction ability on ChartQAPro.
>
> | Model | Minority-correct | Zero-correct |
> | --- | --- | --- |
> | Qwen2.5-VL-72B-Instruct | 45.8% | 1.6% |
> | GPT-4o | 52.0% | 4.0% |

---

> ### Author Response · Authors · 2025-11-21
> **Response to Reviewer 656R (Part 2/6)**
>
> ### **W2 & Q2: Computational efficiency**
>
> We appreciate the reviewer’s suggestion for a more detailed efficiency analysis to support our claims. Below, we report both **API cost and inference time,** which are the most relevant metrics for our setting and directly reflect practical efficiency. We also added Section 4.6 and Appendix C.2 to better demonstrate our cost-efficiency in the revision.
>
> 1. **API cost:**
>
>     As claimed in Section 3.3, **our method mainly improves cost-efficiency through the verdict stage:** the verdict processes multiple draft reasoning paths as prefilled input and outputs only a short answer, rather than invoking large models iteratively for analyzing each image section separately or generating lengthy rationales.
>
>     Thus, **we compare SV against large reasoning model baselines, including GPT-4o, and o1.** Because o1 is very expensive, we randomly sample 1,000 instances from each dataset. The results in Tables 3 and 4 support our claim that SV provides a much better cost–performance trade-off than calling large reasoning models alone:
>
>     - Compared with 4o, SV achieves **6.6-12.0% improvement over GPT-4o while maintaining comparable cost**.
>     - Compared with o1, SV substantially outperforms on InfographicVQA and ChartQAPro, and is comparable on ChartMuseum, **while using only 15–26% of o1’s cost.**
>
>     Table 3: Performance comparison on 1000 random test samples.
>
>     | Method | InfographicVQA | ChartMuseum | ChartQAPro |
>     | --- | --- | --- | --- |
>     | GPT-4o | 76.3 | 42.7 | 51.7 |
>     | o1 | 77.8 | **50.6** | 58.8 |
>     | **SV w/ GPT-4o Verdict** | **88.3** | 49.3 | **63.4** |
>
>     Table 4: Average inference API cost per sample across benchmarks.
>
>     | Method | InfographicVQA | ChartMuseum | ChartQAPro |
>     | --- | --- | --- | --- |
>     | GPT-4o | $0.0038 | $0.0213 | $0.0210 |
>     | o1 | $0.0263  | $0.0663 | $0.0478 |
>     | **SV w/ GPT-4o Verdict** | $0.0068 | $0.0109 | $0.0071 |
> 2. **Inference time:**
>
>     We also measure the average inference time per sample on InfographicVQA using two draft pool configurations, where all draft models are executed in parallel during the draft stage:
>
>     - Main pool (used in our main experiments) that includes two reasoning draft models with chain-of-thought always enabled, so they generate full reasoning even when prompted for direct answers.
>     - Non-reasoning pool that contains only non-reasoning draft models, which can be prompted to provide either direct answers or reasoning paths.
>
>     The results show that SV introduces a **controlled** latency overhead relative to a single draft, but yields a **substantially better** **cost-performance trade-off:**
>
>     - Compared to the best reasoning baseline, SV with the main pool achieves +3.6 ANLS at 1.54× latency.
>     - **Notably, SV with non-reasoning pool outperforms the best reasoning draft while being 2.3× faster.**
>
>     | Method | Performance (ANLS) | Time / sample (seconds) |
>     | --- | --- | --- |
>     | Best reasoning draft | 84.8 | 31.2 |
>     | **SV w/ non-reasoning pool** | 86.3 | 13.8 |
>     | **SV w/ main pool** | 88.4 | 48.0 |

---

> ### Author Response · Authors · 2025-11-21
> **Response to Reviewer 656R (Part 3/6)**
>
> ### **W3 & Q3: Restricted draft model pool and sensitivity to draft model choice**
>
> We appreciate the reviewer’s concern about SV's generalizability across different draft model pools. In the revised version, we study **different pool compositions and model sizes** on InfographicVQA, and add a new subsection “Ablation study on model pool composition” in Appendix E.
>
> Specifically, we evaluate on two more draft pools:
>
> **1. Non-reasoning 7-9B Pool:**
>
> **To test whether SV still works when replacing the two reasoning draft models with faster non-reasoning models**, we construct a 7–9B pool where GLM-4.1V-9B-Thinking and MiMO-VL-7B-RL are replaced by LLaVA-OneVision-1.5-8B and Eagle 2.5-8B, while keeping the remaining three models unchanged. Although these substitutes sacrifice some reasoning capability, they enable faster inference.
>
> The results show that with GPT-4o as verdict, SV surpasses the best draft expert by 4.6% and improves over GPT-4o by 9.8%. **This shows that SV maintains its effectiveness when replacing reasoning models with faster non-reasoning alternatives.**
>
> | Category | Method | Param Size | ANLS |
> | --- | --- | --- | --- |
> | **Closed-source** | GPT-4o | — | 76.5 |
> |  | GPT-4o-mini | — | 67.2 |
> | **Open-source** | Qwen2.5-VL-Instruct | 7B | 79.8 |
> |  | Ovis2.5 | 9B | 81.7 |
> |  | Eagle-2.5 | 8B | 74.5 |
> |  | InternVL3 | 8B | 72.3 |
> |  | LLaVA-OneVision-1.5 | 8B | 70.3 |
> | **Tool-driven** | DeepEyes | 7B | 75.5 |
> |  | Pixel-Reasoner | 7B | 84.0 |
> | **SV (Ours)** | **SV w/ GPT-4o Verdict** | **—** | **86.3** |
> |  | Δ (vs. GPT-4o) | — | +9.8 |
>
> **2. Small 2-4B Pool:**
>
> We further evaluate SV on an even smaller and diverse pool consisting of 2–4B models (Qwen2.5-VL-3B-Instruct, LLaVA-OneVision-1.5-4B, InternVL3.5-4B, Gemma-3-4B, Ovis2.5-2B).
>
> The results show that with GPT-4o as verdict, SV outperforms the best 2–4B draft by 9.5% and yields an 8.0% gain over GPT-4o. **This demonstrates that SV can extract effective collective reasoning even from comparatively weak individual models.**
>
> | Category | Method | Param Size | ANLS |
> | --- | --- | --- | --- |
> | **Closed-source** | GPT-4o | — | 76.5 |
> |  | GPT-4o-mini | — | 67.2 |
> | **Open-source** | Qwen2.5-VL-Instruct | 3B | 64.9 |
> |  | Ovis2.5  | 2B | 75.0 |
> |  | InternVL3.5 | 4B | 74.4 |
> |  | LLaVA-OneVision-1.5 | 4B | 67.1 |
> |  | Gemma 3 | 4B | 36.0 |
> | **Tool-driven** | DeepEyes | 7B | 75.5 |
> |  | Pixel-Reasoner | 7B | 84.0 |
> | **SV (Ours)** | **SV w/ GPT-4o Verdict** | **—** | **84.5** |
> |  | Δ (vs. GPT-4o) | — | +8.0 |
>
> Across these settings, the draft pools differ in size and architecture, but SV consistently delivers gains over the baselines. These results indicate that **SV is not overly sensitive to a specific draft pool: it robustly leverages complementary reasoning from a variety of model families and scales, while maintaining cost efficiency.**

---

> ### Author Response · Authors · 2025-11-21
> **Response to Reviewer 656R (Part 4/6)**
>
> ### **W4 & Q4: Failure case analysis**
>
> We appreciate this important suggestion to analyze SV's failure modes. We have conducted failure analysis and added the results to Appendix J in the revised manuscript, focusing on the representative benchmark InfographicVQA with GPT-4o as verdict.
>
> To identify the failure cases, we **focus on instances where the majority of drafts are correct but SV produces wrong answers**, as these reveal when the consensus mechanism or verdict synthesis fails despite having correct information available. Under strict exact-match scoring, such cases account for 3.98% of all “majority-correct” examples. After normalizing answer formats (e.g., mapping “Feb 11” vs. “February 11”), many of these are actually correct; the true failure rate drops to **1.34%** (34 cases).
>
> Among these 34 failure cases, two dominant patterns emerge as their proportions are shown in the following table:
>
> 1. **Fine-grained extraction (52.9%):**
>
>     Errors arise from extracting tiny numbers or small text on very tall infographics. When draft models extract these details correctly, GPT-4o as verdict sometimes **cannot verify them** due to its tile-based high-resolution processing, which can lose subtle information. In these cases, the verdict’s visual perception becomes the bottleneck.
>
> 2. **Color matching (35.3%):**
>
>     These questions require matching a colored legend or region to the correct label. **We find that both the drafts and the verdict frequently struggle with color discrimination and grounding:**
>
>     - Some draft models produce vague reasoning, often “guessing” when they cannot reliably identify the target color.
>     - GPT-4o can be misled by fluent but incorrect reasoning. In 58.3% of such cases, it follows the answer from the draft model with the longest reasoning trajectory, suggesting a bias towards verbose explanations when visual cues are weak.
>
>     These failures reflect a broader limitation of current VLMs in precise color reasoning, which SV cannot fully overcome when all experts are visually uncertain.
>
>
> | Failure mode | Proportion |
> | --- | --- |
> | Fine-grained extraction | 52.9% |
> | Color matching | 35.3% |
> | Other | 11.8% |
>
> These analyses clarify SV’s limitations: **(i) it inherits current VLM weaknesses in color reasoning and fine-grained perception, and (ii) when the visual evidence provided by the drafts is ambiguous, the verdict can over-weight verbose but incorrect explanations.**

---

> ### Author Response · Authors · 2025-11-21
> **Response to Reviewer 656R (Part 5/6)**
>
> ### **W5 & Q5: Comparison to more advanced ensemble methods**
>
> We appreciate the suggestion to compare SV against more advanced ensembling strategies. In a multimodal generation setting, sophisticated aggregation can naturally be realized via the LMM-as-a-Judge paradigm, where a learned critic takes multiple candidate reasoning paths and scores or ranks them to select the best answer.
>
> In the revised version, we therefore adopt **LLaVA-Critic [1]** as an additional baseline. It is trained as a **generalist multimodal evaluator** that jointly uses visual evidence and textual reasoning to compare answers across a wide range of vision–language tasks, making it a stronger and more appropriate ensemble-style baseline for our information-intensive visual reasoning setting than classical text-only ensemble methods. Building on this, we implement comprehensive experiments with **LLaVA-Critic-7B** and **LLaVA-Critic-72B**, and add the results in Section 4.5 and Appendix G.
>
> 1. **Experimental Setup:**
>     1. We strictly follow LLaVA-Critic's standardized evaluation template and test two judging modes:
>         - Pointwise scoring: the judge assigns scores to each candidate and selects the highest
>         - Pairwise ranking: the judge compares candidates and selects a winner (i.e., in our scenario, it comthe pares first two drafts, then compares the winner vs. the third draft)
>     2. To ensure a comprehensive comparison, we evaluate LLaVA-Critic in two settings:
>         1. Judge the same 3 draft experts’ reasoning selected in SV (direct replacement of SV's verdict)
>         2. Judge the 5 candidate models’ reasoning
> 2. **Results on LLaVA-Critic-7B judging 3 draft experts:**
>
>     We observe that SV is substantially more accurate than LLaVA-Critic-7B on all three benchmarks (+4.9-11.9%). Qualitatively, LLaVA-Critic is limited to selecting a single candidate and is often influenced by surface form (e.g., style of reasoning, length, repetitiveness), whereas **SV’s verdict is to synthesize and cross-check factual cues across multiple trajectories, making it less sensitive to stylistic artifacts and better at recovering correct answers from partial evidence.**
>
>     | Method | InfographicVQA | ChartMuseum | ChartQAPro |
>     | --- | --- | --- | --- |
>     | LLaVA-Critic-7B (pointwise) | 83.5 | 40.1 | 52.4 |
>     | LLaVA-Critic-7B (pairwise) | 81.4 | 38.9 | 52.1 |
>     | **SV w/ GPT-4o Verdict** | **88.4** | **49.3** | **64.0** |
> 3. **Results on LLaVA-Critic-7B judging 5 candidate models:**
>
>     We evaluate on 1,000 randomly sampled examples per dataset for quick comparison. It shows that the 7B critic remains substantially below SV, confirming that simply adding more candidates to this judge does not close the gap.
>
>     | Method | InfographicVQA | ChartMuseum | ChartQAPro |
>     | --- | --- | --- | --- |
>     | LLaVA-Critic-7B (pointwise) | 82.3 | 34.6 | 56.5 |
>     | **SV w/ GPT-4o Verdict** | **88.3** | **49.3** | **63.4** |
> 4. **Results on LLaVA-Critic-72B judging 3 draft experts:**
>
>     We further repeat the (2) experiment with LLaVA-Critic-72B as a stronger judge (on 1,000 random examples per dataset). LLaVA-Critic-72B is still notably weaker than SV on all three benchmarks.
>
>     | Method | InfographicVQA | ChartMuseum | ChartQAPro |
>     | --- | --- | --- | --- |
>     | LLaVA-Critic-72B (pointwise) | 82.6 | 40.4 | 53.5 |
>     | LLaVA-Critic-72B (pairwise) | 83.1 | 43.0 | 55.8 |
>     | **SV w/ GPT-4o Verdict** | **88.3** | **49.3** | **63.4** |
>
> In summary, SV consistently outperforms standard LMM-as-a-Judge frameworks (LLaVA-Critic 7B and 72B). This demonstrates that SV provides a more effective way than selection-based judging for information-intensive visual reasoning.
>
> [1] LLaVA-Critic: Learning to Evaluate Multimodal Models, CVPR 2025

---

> ### Author Response · Authors · 2025-11-21
> **Response to Reviewer 656R (Part 6/6)**
>
> ### **W6 & Q6: Generalization to visual reasoning benchmarks**
>
> We appreciate this question about SV's generalizability across diverse visual reasoning tasks. To address this, we additionally evaluated SV on **two different types of visual reasoning tasks** and added the results in Section 4 in the revision:
>
> - **TallyQA-Complex** (1,000 randomly sampled complex questions): open-ended counting that requires reasoning over object relations and attributes.
> - **MathVista** (official testmini set including 1000 samples): mathematical reasoning in visual contexts, emphasizing fine-grained understanding and compositional reasoning.
>
> We use the same draft pool as in the main experiments and keep GPT-4o as the verdict. The results indicate that **SV provides consistent gains on counting and mathematical visual reasoning tasks, supporting its generalization beyond the benchmarks evaluated in the main paper:**
>
> - On TallyQA-Complex, SV improves over the best draft expert by +3.0% and over GPT-4o by +1.5%.
> - On MathVista, SV outperforms the best draft expert by +2.6% and substantially surpasses GPT-4o by +17.8%.
>
> | Category | Model | Param Size | TallyQA-Complex | MathVista |
> | --- | --- | --- | --- | --- |
> | **Closed-source** | GPT-4o | — | 75.4 | 65.1 |
> | **Open-source** | Qwen2.5-VL-Instruct | 7B | 72.4 | 68.2 |
> |  | MiMO-VL-RL | 7B | 72.0 | 80.3 |
> |  | InternVL3 | 8B | 70.1 | 72.7 |
> |  | GLM-4.1V-Thinking | 9B | 73.9 | 78.9 |
> |  | Ovis2.5 | 9B | 71.9 | 77.3 |
> | **SV (Ours)** | **SV w/ GPT-4o verdict** | — | **76.9** | **82.9** |
> |  | Δ (vs. GPT-4o) | — | +1.5 | +17.8 |

---

> > ### Comment · Reviewer_656R · 2025-11-26
> >
> > Thanks for the response. My concerns are well addressed and included in the revision. I will increase my rating accordingly.

---

> > > ### Author Response · Authors · 2025-11-27
> > >
> > > Thank you for the positive feedback and for raising the score!

---

### Official Review · Reviewer_Cnwz · 2025-11-01

**Soundness:** 3
**Presentation:** 3
**Contribution:** 3
**Rating:** 6
**Confidence:** 3

**Summary:**

This paper addresses Large Vision-Language Models’ struggles with information-intensive images—difficulty localizing critical cues in dense layouts and multi-hop reasoning. It proposes Speculative Verdict (SV), a training-free framework combining lightweight draft experts and a large verdict model. Small VLMs generate diverse reasoning paths in the draft stage; a strong VLM synthesizes these for final answers in the verdict stage. SV adds a consensus expert selection to forward only high-agreement paths. Experiments show SV gains on benchmarks like InfographicVQA, achieving error correction and cost-efficiency vs. large models.

**Strengths:**

1) This paper accurately pinpoints VLMs’ core flaws in information-intensive images—poor dense cue localization and error-prone multi-hop reasoning—and clarifies limitations of existing solutions, ensuring relevance.
2) The “Draft-Verdict” two-stage structure (lightweight experts for coverage + large VLM for synthesis) and consensus selection balance accuracy and efficiency, with clear alignment to solving target challenges.
3) Experiments on diverse benchmarks (InfographicVQA, HR-Bench 4K) and comparisons with various baselines, plus error correction data (47-53%), fully validate performance and cost-efficiency.

**Weaknesses:**

1) How about the comparison of the proposed method with specialized models?
2) The inference speed is not presented in the experiments section. Does it add much computation cost to the baseline method thus slow down the inference1 speed, and if yes, could you give the speed?
3) On HR-Bench 4K, SV w/ GPT-4o Verdict performs worse than SV w/ Qwen2.5-VL-72B-Instruct Verdict, and even worse than several Open-source VLMs, please explain why?

**Questions:**

See Weaknesses.

---

> ### Author Response · Authors · 2025-11-21
> **Response to Reviewer Cnwz (Part 1/3)**
>
> We thank you for the thoughtful feedback and positive review! Below, we address each question in detail, and all revisions are marked in blue in the revised manuscript.
>
> ### **W1: Comparison with specialized models**
>
> We appreciate the question about how SV compares to task-specialized models. In the revision, we added **PixelReasoner [1]** as a key baseline, and report it alongside the existing tool-driven baseline **DeepEyes [2]** in our main results table. Briefly, both are well aligned with our setting:
>
> - PixelReasoner is trained for information-rich visual reasoning. It equips a VLM with visual operations (e.g., zoom-in) and uses a two-stage post-training pipeline, which is instruction tuning on synthesized traces followed by curiosity-driven RL with queries including InfographicVQA.
> - DeepEyes is a tool-augmented framework trained on fine-grained, chart, and reasoning data, interleaving zoom-in tools with the reasoning process to improve local perception.
>
> We evaluate all methods on three information-intensive benchmarks and observe that **SV consistently outperforms both specialized baselines by a clear margin**, as shown in the following table. The tool-driven models benefit from zoom-in operations when they are correctly triggered, but their tool calls are often **under-triggered or mis-triggered** on information-intensive images, which limits their effectiveness (see Appendix K for a detailed failure mode analysis of DeepEyes). In contrast, SV synthesizes complementary reasoning from diverse VLM experts under a strong verdict model, leading to more robust performance across different layouts and question types.
>
> | Method | InfographicVQA | ChartMuseum | ChartQAPro |
> | --- | --- | --- | --- |
> | Pixel-Reasoner | 84.0† | 25.9 | 39.3 |
> | DeepEyes | 75.7 | 28.0 | 48.7 |
> | **SV w/ GPT-4o Verdict** | **88.4** | **49.3** | **64.0** |
>
> † Result is reported from its original paper; other numbers are obtained from our reproduction under the same setting.
>
> [1] Pixel Reasoner: Incentivizing Pixel-Space Reasoning with Curiosity-Driven Reinforcement Learning, NeurIPS 2025
>
> [2] DeepEyes: Incentivizing" Thinking with Images" via Reinforcement Learning. arXiv preprint arXiv:2505.14362.

---

> ### Author Response · Authors · 2025-11-21
> **Response to Reviewer Cnwz (Part 2/3)**
>
> ### **W2: Inference speed and computational cost**
>
> We appreciate this important practical concern. Below, we report both **inference time** and **API cost** for a comprehensive efficiency analysis. Our cost-efficiency gains mainly come from the verdict stage because SV uses draft reasoning as prefilling input and generates only short answers for once, detailed in the second API cost section below. We also added Section 4.6 and Appendix C.2 to better demonstrate our cost-efficiency in the revision.
>
> 1. **Inference time:**
>
>     We measure the average inference time per sample on InfographicVQA using two draft pool configurations, where all draft models are executed in parallel during the draft stage:
>
>     - Main pool (used in our main experiments) that includes two reasoning draft models with chain-of-thought always enabled, so they generate full reasoning even when prompted for direct answers.
>     - Non-reasoning pool that contains only non-reasoning draft models, which can be prompted to provide either direct answers or reasoning paths.
>
>     The results show that SV introduces a **controlled** latency overhead relative to a single draft, but yields a **substantially better cost-performance trade-off:**
>
>     - Compared to the best reasoning baseline, SV with the main pool achieves +3.6 ANLS at 1.54× latency.
>     - **Notably, SV with non-reasoning pool outperforms the best reasoning draft while being 2.3× faster.**
>
>     | Method | Performance (ANLS) | Time / sample (seconds) |
>     | --- | --- | --- |
>     | Best reasoning draft | 84.8 | 31.2 |
>     | **SV w/ non-reasoning pool** | 86.3 | 13.8 |
>     | **SV w/ main pool** | 88.4 | 48.0 |
> 2. **API cost:**
>
>     As claimed in Section 3.3, **our method mainly improves cost-efficiency through the verdict stage:** the verdict processes multiple draft reasoning paths as prefilled input and outputs only a short answer, rather than invoking large models iteratively for analyzing each image section separately or generating lengthy rationales.
>
>     Thus, we compare SV against large reasoning model baselines, including GPT-4o, and o1. Because o1 is very expensive, we randomly sample 1,000 instances from each dataset. The results in Tables 1 and 2 show that:
>
>     - Compared with 4o, SV achieves **6.6-12.0% improvement over GPT-4o while maintaining comparable cost**.
>     - Compared with o1, SV substantially outperforms on InfographicVQA and ChartQAPro, and is comparable on ChartMuseum, **while using only 15–26% of o1’s cost.**
>
>     Table 1: Performance comparison on 1000 random test samples.
>
>     | Method | InfographicVQA | ChartMuseum | ChartQAPro |
>     | --- | --- | --- | --- |
>     | GPT-4o | 76.3 | 42.7 | 51.7 |
>     | o1 | 77.8 | **50.6** | 58.8 |
>     | **SV w/ GPT-4o Verdict** | **88.3** | 49.3 | **63.4** |
>
>     Table 2: Average inference API cost per sample across benchmarks.
>
>     | Method | InfographicVQA | ChartMuseum | ChartQAPro |
>     | --- | --- | --- | --- |
>     | GPT-4o | $0.0038 | $0.0213 | $0.0210 |
>     | o1 | $0.0263  | $0.0663 | $0.0478 |
>     | **SV w/ GPT-4o Verdict** | $0.0068 | $0.0109 | $0.0071 |

---

> ### Author Response · Authors · 2025-11-21
> **Response to Reviewer Cnwz (Part 3/3)**
>
> ### **W3: Weaker Performance of SV w/ GPT-4o verdict on HR-Bench 4K**
>
> We thank the reviewer for pointing out this behavior on HR-Bench 4K. **This is a benchmark-specific phenomenon and demonstrates SV's task-adaptive verdict selection**, which we briefly mentioned in Section 4.3.
>
> HR-Bench 4K is designed to test **4K-resolution perception**: its images have an average resolution of ~4032, and many questions hinge on very fine-grained local details. In this setting, the verdict model should accurately perceive subtle visual cues in the drafts’ reasoning to make reliable verifications and corrections.
>
> However, GPT-4o currently processes images via a tile-based pipeline with downsampling (e.g., resizing to at most a 2048×2048 frame and representing content via 512×512 tiles), which leads to noticeable information loss on 4K images. This makes it harder for GPT-4o to reliably distinguish small objects, fine text, or tiny color differences, even when the draft models’ reasoning is strong.
>
> By contrast, Qwen2.5-VL-72B-Instruct employs dynamic resolution processing with window attention, converting images of varying sizes into sequences of tokens with corresponding lengths, which preserves more spatial detail. As a result, it has stronger localization and fine-grained perception on 4K images, which directly benefits its role as a verdict model on HR-Bench 4K.

---

> > ### Author Response · Authors · 2025-11-27
> > **Response to Reviewer Cnwz**
> >
> > Dear Reviewer Cnwz,
> >
> > We sincerely appreciate your time and valuable feedback. As the discussion period is approaching its end **in less than a week**, we want to confirm whether our responses have effectively addressed your concerns. We have added detailed experiments, analyses, and comparisons in both the rebuttal and revised manuscript.
> >
> > We would be happy to address any additional points if needed. Thank you for your time and engagement.
> >
> > Best regards,
> >
> > Authors

---

### Official Review · Reviewer_65ys · 2025-11-02

**Soundness:** 2
**Presentation:** 3
**Contribution:** 2
**Rating:** 4
**Confidence:** 5

**Summary:**

This paper presents a prompting framework for high-resolution image reasoning tasks, where small LVLMs generate draft reasoning trajectories and answers, and a large verdict model incorporates all reasoning and produces the final answers. The reasoning trajectories are selected based on the consensus score (i.e., the absolute difference between the model's own answer and the answer generated by the other model).  Experimental results show the framework achieves better results on high-resolution/dense-layout benchmarks such as InfoVQA and HRBench.

**Strengths:**

- The investigated problem of solving dense-layout image reasoning tasks using ensemble learning is of great practical value.
- The performance is promising, surpassing tool-based methods such as DeepEyes.

**Weaknesses:**

- Unclear Connection to Speculative Decoding
The paper's framing as "speculative decoding" is confusing. Traditional speculative decoding aims at inference acceleration, whereas this work operates more as an LLM-as-a-Judge paradigm where candidate answers are evaluated by a verdict model. The paper lacks discussion and comparison with existing judging frameworks (e.g., [1, 2]), which weakens its positioning within the literature.

- Limited Technical Contribution:
Viewing this work through the LLM-as-a-Judge lens, the technical novelty appears limited beyond modifying the aggregation process with consensus scores. Despite the ablation in Figure 7, several critical aspects remain underexplored:
  - What is the distribution of agreement/disagreement among models?
  - How does normalization affect the results? Could overconfident models skew the consensus?
  - Since answers are generated with reasoning trajectories, does estimating NLL on answers alone introduce inaccuracy due to off-policy estimation?

- Insufficient Motivation and Analysis:
The paper would benefit from deeper investigation to strengthen its claims:
  - A detailed analysis of reasoning trajectory patterns across different models
  - Exploration of whether using a smaller model as the verdict would yield similar performance gains
  - Token efficiency comparison against standard LLM-as-a-Judge frameworks, since both approaches can leverage prefilling



[1] LLaVA-Critic: Learning to Evaluate Multimodal Models, CVPR 2025
[2] VL-RewardBench: A Challenging Benchmark for Vision-Language Generative Reward Models, CVPR 2025

**Questions:**

## Questions

- The reasoning trajectory analysis is interesting in Figure 1 and Section 3.2. I am wondering if there is any quantitative analysis of reasoning type distribution between draft models and their influence, e.g., which models are more complementary?

- Why is the ablation study about the verdict scale only conducted on a subset of InfoVQA?

- Will some of the small reasoning models produce a long reasoning trajectory, and then the final prefill stage exceeds the verdict model's length limit?
## Format
- Table 7 -> Figure 7.

---

> ### Author Response · Authors · 2025-11-21
> **Response to Reviewer 65ys (Part 1/7)**
>
> We thank the reviewer for the careful reading and constructive feedback! We have revised the paper to clarify our conceptual positioning, to strengthen the technical analysis, and to improve formatting. We address each concern in detail below, and all revisions are marked in blue in the revised manuscript.
>
> ### **W1: Connection to speculative decoding and LMM-as-a-Judge**
>
> We appreciate the reviewer’s insightful comment on the positioning of our framework. We agree that our framework has strong connections to the LMM-as-a-Judge paradigm and thank the reviewer for encouraging us to better situate our work in this line of research.
>
> In the revised manuscript, we add a paragraph titled **“LMM-as-a-Judge”** in the Related Work (Section 2), where we discuss representative judging frameworks such as LLaVA-Critic [1], and clarify how they use large multimodal models to score, rank, or select candidate answers. We also make our own positioning explicit: in SV, the verdict model is used as an **off-the-shelf multimodal judge** that **filters informative cues from diverse draft experts and synthesizes a final answer** for information-intensive images. Additionally, regarding **cost efficiency compared with standard judging frameworks**, we provide a detailed token efficiency analysis in our response to Weakness 3.3, which shows that SV achieves favorable cost–performance trade-offs against the LMM-as-a-Judge baseline.
>
> Regarding the **connection to speculative decoding**, our intention is to draw on its **high-level principle**, not to claim that SV is a token-level speculative decoding algorithm or that our primary goal is inference acceleration. As we emphasized at the beginning of Section 3 in the paper, speculative decoding and SV share the same core idea:
>
> > “…draft models expand coverage quickly, while the verifier ensures correctness. Although this idea has been mainly applied to accelerate text generation, its high-level principle is also well-suited for information-intensive multimodal reasoning.”
> >
>
> We hope this refined framing addresses the reviewer’s concern, and we thank the reviewer for helping us to sharpen our positioning.
>
> [1] LLaVA-Critic: Learning to Evaluate Multimodal Models, CVPR 2025

---

> ### Author Response · Authors · 2025-11-21
> **Response to Reviewer 65ys (Part 2/7)**
>
> ### **W2: Further analysis and ablations of the consensus mechanism**
>
> We appreciate the reviewer's suggestions for deeper technical analysis! We addressed each aspect below and added relevant discussion in Appendix F.
>
> ---
>
> ### **W2.1: Distribution of agreement/disagreement among models**
>
> To understand how our consensus mechanism selects draft experts, we analyze the selection frequency of each model across benchmarks. Table 1 shows the proportion of instances where each model is selected. The results show a **diverse participation pattern:** All models are selected in 38.6-84.7% of instances, with no model dominating or being marginalized. This indicates that (i) different models specialize on different subsets of questions, and (ii) the consensus mechanism indeed leverages **complementary agreement and disagreement** across experts rather than collapsing to a single “always selected” model. This complementary phenomenon is further analyzed in response to Weakness 3.1 & Q1.
>
> Table 1: Selection frequency of each model across benchmarks.
>
> | Model | InfographicVQA | ChartQAPro |
> | --- | --- | --- |
> | Qwen2.5-VL-7B-Instruct | 84.7% | 77.7% |
> | GLM-4.1V-9B-Thinking | 38.6% | 69.3% |
> | MiMO-VL-7B-RL | 54.8% | 56.8% |
> | InternVL3-8B | 73.4% | 53.1% |
> | Ovis2.5-9B | 48.5% | 43.1% |
>
> ---
>
> ### **W2.2: Effect of normalization on consensus score**
>
> We further ablate the effect of normalization, which was originally motivated by the calibration gap across models. Different VLMs produce perplexity scores on systematically different scales due to training and tokenization differences. In our draft pool, for example, Qwen2.5-VL and GLM-4.1V-Thinking tend to output larger perplexity values. Without normalization, these scale differences cause models with larger-magnitude scores to **dominate the consensus, even when they do not actually agree more often.**
>
> We study this on **two draft pools** on InfographicVQA: 1) **Main pool (used in our primary experiments)** with mixed reasoning and non-reasoning models; 2) **Additional pool (all non-reasoning models)**.
>
> 1. For the main pool, Tables 2 and 3 summarize the effect of normalization:
>     - **Normalization restores diversity:** Without normalization, Qwen and GLM are selected in 99.9-100% of instances, turning the top-k pool into a near-fixed subset. With normalization, selection becomes balanced (38.6-84.7% range in Table 1), allowing all models to contribute based on actual agreement.
>     - **Performance:** On ChartQAPro, normalization brings a clear gain of +4.6 points, while on InfographicVQA the performance is comparable. In the latter case, the unnormalized setting happens to pick a strong pairing of reasoning models, but this is accidental rather than principled, validated by the following experiment on the additional pool.
>
>     Table 2: Ablation on normalization on the main draft pool.
>
>     | Variant | InfographicVQA | ChartQAPro |
>     | --- | --- | --- |
>     | SV w/o normalization | 88.9 | 59.4 |
>     | SV | 88.4 | 64.0 |
>
>     Table 3: Selection frequency of each model without normalization.
>
>     | Model | InfographicVQA | ChartQAPro |
>     | --- | --- | --- |
>     | Qwen2.5-VL-7B-Instruct | 99.9% | 100.0% |
>     | GLM-4.1V-9B-Thinking | 77.1% | 99.9% |
>     | MiMO-VL-7B-RL | 57.0% | 71.5% |
>     | InternVL3-8B | 47.0% | 21.4% |
>     | Ovis2.5-9B | 19.1% | 2.2% |
> 2. **To verify that this is not specific to a particular pool**, we repeat the ablation on an additional pool consisting only of non-reasoning models on InfographicVQA. As shown in Tables 4 and 5, **normalization provides +3.7% improvement and produces a more diverse selection pattern.**
>
>     Table 4: Ablation on normalization on the additional draft pool.
>
>     | Variant | InfographicVQA |
>     | --- | --- |
>     | SV w/o normalization | 84.6 |
>     | SV | 86.3 |
>
>     Table 5: Selection frequencies on the additional draft pool.
>
>     | Model | InfographicVQA (w/ norm.) | InfographicVQA (w/o norm.) |
>     | --- | --- | --- |
>     | Qwen2.5-VL-7B-Instruct | 87.3% | 99.9% |
>     | Eagle2.5-8B | 65.4% | 98.0% |
>     | LLaVA-OneVision-1.5-8B | 28.6% | 79.3% |
>     | InternVL3-8B | 74.5% | 16.7% |
>     | Ovis2.5-9B | 44.2% | 6.11% |

---

> ### Author Response · Authors · 2025-11-21
> **Response to Reviewer 65ys (Part 3/7)**
>
> ### **W2.3: NLL Estimation on Answers vs. Full Trajectories**
>
> We appreciate the reviewer’s concern that computing NLL only on the final answers might introduce inaccuracy, since the answers are generated together with reasoning trajectories. To investigate this, we conducted ablations comparing two NLL estimation strategies:
>
> - Answer-only NLL (SV): compute perplexity only on answer tokens produced by QA.
> - Full-trajectory NLL: compute perplexity on the entire reasoning trajectory.
>
> The results show that the two variants achieve very similar performance on both benchmarks, suggesting that **any off-policy bias introduced by answer-only scoring is negligible for our task.** At the same time, scoring only the answers is more **computationally efficient**, as it avoids computing NLL over long reasoning traces with many extra tokens; **answer-only NLL also provides a cleaner signal** by avoiding noise from diverse reasoning styles across models. Given the above trade-off, we adopt the answer-only NLL variant in the main experiments and now include this ablation in Appendix F.3 for clarity.
>
> | Scoring variant | InfographicVQA | ChartQAPro |
> | --- | --- | --- |
> | Answer+reasoning NLL | 87.9 | 64.3 |
> | Answer-only NLL (SV) | 88.4 | 64.0 |

---

> ### Author Response · Authors · 2025-11-21
> **Response to Reviewer 65ys (Part 4/7)**
>
> ### **W3.1 & Q1: Quantitative analysis of reasoning trajectories and model complementarity**
>
> We thank the reviewer for the suggestion! In the revised version, we add a detailed quantitative analysis of draft complementarity in Appendix I.
>
> To understand complementarity, we focus on **minority-correct recovery cases** **where only one of the three selected experts is correct and SV subsequently recovers the correct answer.** These cases are most informative because they reveal how specific models provide unique, correct information and how others behave to help the verdict distinguish cues. **We analyzed 50 such cases from InfographicVQA and manually categorized five major reasoning bottlenecks:**
>
> | Reasoning Bottleneck | Description | Frequency (%) |
> | --- | --- | --- |
> | **Extraction** | Locating and reading fine-grained text/numbers | 50 |
> | **Color matching** | Distinguishing and matching colors in legends/charts | 18 |
> | **Global scan** | Aggregating information across entire image | 16 |
> | **Localization** | Finding specific query-relevant regions | 10 |
> | **Numerical comparison** | Comparing numerical values | 4 |
>
> The table below shows each model's success rate on the 50 minority-correct recovery cases, broken down by reasoning bottleneck. **We observe a clear division of labor where no single model dominates across all reasoning types, and each model has distinct strengths:**
>
> - GLM-4.1V-9B-Thinking and MiMO-VL-7B-RL excel at fine-grained extraction (58% and 70% success), and MiMO-VL-7B-RL is also particularly strong on global scan (75% success). This is because as reasoning-oriented models, their detailed step-by-step generation allows them to iteratively verify extracted values and cross-check multiple regions.
> - Ovis2.5-9B and GLM-4.1V-9B-Thinking are strong on color matching (57% and 60% success), handling legend-color alignment better than other models.
> - Qwen dominates localization (80%) and numerical comparison (100%). We observe that other models often fail due to keyword comprehension errors. Then, Qwen's correct understanding makes its answers stand out clearly for the verdict to identify, even when other models provide detailed but misdirected reasoning.
>
> | Reasoning Type | Qwen2.5-VL-7B-Instruct | GLM-4.1V-9B-Thinking | MiMO-VL-7B-RL | Ovis2.5-9B | InternVL3-8B |
> | --- | --- | --- | --- | --- | --- |
> | **Extraction** | 15% | **58%** | **70%** | 38% | 15% |
> | **Color matching** | 0% | **60%** | 33% | **57%** | 25% |
> | **Localization** | **80%** | 0% | 0% | 33% | 0% |
> | **Global scan** | 0% | 33% | **75%** | 20% | 0% |
> | **Numerical comparison** | **100%** | 0% | 0% | 0% | 0% |
>
> The table below reports the **most frequent model combination among all 243 minority-correct recovery cases** on InfographicVQA. The top combinations are Qwen + Ovis + InternVL, Qwen + MiMO + InternVL, and Qwen + GLM + Ovis, which correspond to two complementarity patterns:
>
> **(i) Balanced visual skills (Qwen + Ovis + InternVL):** All three models offer comparable extraction ability, but with specialized strengths. This combination works well when questions require diverse visual skills rather than deep extraction reasoning.
>
> **(ii) Extraction-focused pairings (Qwen + MiMO + InternVL, Qwen + GLM + Ovis):** They appear predominantly in extraction-intensive cases where fine-grained information is difficult to locate. The reasoning models supply the correct detailed values through step-by-step verification, whereas the non-reasoning models contribute robust localization and partial cues that the verdict model leverages for cross-verification.
>
> | Rank | Model Combination | Frequency |
> | --- | --- | --- |
> | 1 | Qwen + Ovis + InternVL | 33.3% |
> | 2 | Qwen + MiMO + InternVL | 14% |
> | 3 | Qwen + GLM + Ovis | 11.1% |
>
> Overall, this quantitative analysis supports our claim that SV does not simply average similar models: it exploits complementary reasoning strengths across draft experts, and the minority-correct recoveries we observe are closely associated with diverse and specialized reasoning trajectories across models.

---

> ### Author Response · Authors · 2025-11-21
> **Response to Reviewer 65ys (Part 5/7)**
>
> ### **W3.2 & Q2: Using smaller models as the verdict**
>
> We appreciate this question about whether a smaller model could serve as an effective verdict. In the original paper, we reported an ablation on a 1000-example subset of InfographicVQA for quick comparison (GLM-4.1V-9B-Thinking runs relatively slowly as a reasoning model), and this scale was sufficient to reveal the trend. Based on the reviewer’s feedback, we extend the experiments to the **full test sets of all three benchmarks** and to **multiple smaller verdict models**.
>
> Concretely, we keep the draft pool fixed and replace GPT-4o as the verdict with three strong smaller models: GLM-4.1V-9B-Thinking (reasoning-oriented), MiMO-VL-7B-RL (reasoning-oriented), and Qwen2.5-VL-7B-Instruct (high-performing non-reasoning model). The results in Tables 1 and 2 show that:
>
> - **Smaller verdict models underperform the large verdict across all datasets.**
> - Reasoning models (i.e., GLM-4.1V-9B-Thinking and MiMO-VL-7B-RL) **produce 60-200x more output tokens, yet still underperform SV.** This shows that replacing the large verdict with a smaller reasoning model does not yield a better cost–performance trade-off.
> - Under similar token costs, Qwen2.5-VL-7B-Instruct delivers evident weaker performance than SV.
>
> Table 1: Performance with different verdict models.
>
> | Verdict | InfographicVQA | ChartMuseum | ChartQAPro |
> | --- | --- | --- | --- |
> | GLM-4.1V-9B-Thinking | 84.7 | 48.0 | 62.6 |
> | MiMO-VL-7B-RL | 85.4 | 46.9  | 60.3 |
> | Qwen2.5-VL-7B-Instruct | 84.1  | 47.1 | 57.2 |
> | **GPT-4o (SV)** | **88.4** | **49.3** | **64.0** |
>
> Table 2: Average verdict output tokens per sample across benchmarks.
>
> | Verdict | InfographicVQA | ChartMuseum | ChartQAPro |
> | --- | --- | --- | --- |
> | GLM-4.1V-9B-Thinking | 272.1 | 604.5 | 440.5 |
> | MiMO-VL-7B-RL | 183.2 | 368.2 | 378.1 |
> | Qwen2.5-VL-7B-Instruct | 3.3 | 3.4 | 2.8 |
> | **GPT-4o (SV)** | 2.7 | 3.0 | 2.2 |
>
> These extended experiments confirm that using a smaller verdict model does not reproduce the gains of SV with a strong verdict model, validating that a large, capable verdict is essential for effective synthesis. We have added these results in Section 4.5 and Appendix C.2 to clarify this point.

---

> ### Author Response · Authors · 2025-11-21
> **Response to Reviewer 65ys (Part 6/7)**
>
> ### **W3.3: Token efficiency comparison against standard LLM-as-a-Judge frameworks**
>
> We appreciate this important concern about computational efficiency compared to standard judging frameworks. To provide a direct comparison, we implemented comprehensive experiments using **LLaVA-Critic** [1] as our baseline, testing both 7B and 72B model scales. We have also added the results in Section 4.5 and Appendix G.
>
> 1. **Experimental Setup:**
>     1. We strictly follow LLaVA-Critic's standardized evaluation template and test two judging modes:
>         - Pointwise scoring: the judge assigns scores to each candidate and selects the highest
>         - Pairwise ranking: the judge compares candidates and selects a winner (i.e., in our scenario, it compares the first two drafts, then compares the winner vs. the third draft)
>     2. To ensure a comprehensive comparison, we evaluate LLaVA-Critic in two settings:
>         1. Judge the same 3 draft experts’ reasoning selected in SV (direct replacement of SV's verdict)
>         2. Judge the 5 candidate models’ reasoning
> 2. **Results on LLaVA-Critic-7B judging 3 draft experts:**
>
>     We observe that **SV achieves a strictly better cost–performance trade-off than LLaVA-Critic-7B** acting as a judge over the same drafts:
>
>     - SV is substantially more accurate than LLaVA-Critic-7B on all three benchmarks (+4.9-11.9%).
>     - Token usage is comparable or lower for SV: it uses fewer verdict tokens than both pointwise and pairwise LLaVA-Critic on InfographicVQA and ChartMuseum, and lies between the two variants on ChartQAPro while still achieving a much higher accuracy.
>     - Qualitatively, LLaVA-Critic is limited to selecting a single candidate and is often influenced by surface form (e.g., style of reasoning, length, repetitiveness), whereas **SV’s verdict is to synthesize and cross-check factual cues across multiple trajectories, making it less sensitive to stylistic artifacts and better at recovering correct answers from partial evidence.**
>
>     Table 1: Performance with SV vs. LLaVA-Critic-7B as judge.
>
>     | Method | InfographicVQA | ChartMuseum | ChartQAPro |
>     | --- | --- | --- | --- |
>     | LLaVA-Critic-7B (pointwise) | 83.5 | 40.1 | 52.4 |
>     | LLaVA-Critic-7B (pairwise) | 81.4 | 38.9 | 52.1 |
>     | **SV w/ GPT-4o Verdict** | **88.4** | **49.3** | **64.0** |
>
>     Table 2: Average judge/verdict tokens per sample, including input and output tokens.
>
>     | Method | InfographicVQA | ChartMuseum | ChartQAPro |
>     | --- | --- | --- | --- |
>     | LLaVA-Critic-7B (pointwise) | 1053.3 | 3723.7 | 1290.7 |
>     | LLaVA-Critic-7B (pairwise) | 1342.9 | 4689.4 | 1759.1 |
>     | **SV w/ GPT-4o Verdict** | 701.7 | 3122.3 | 1441.7 |
> 3. **Results on LLaVA-Critic-7B judging 5 candidate models:**
>
>     We evaluate on 1000 randomly sampled examples per dataset for quick comparison. It shows that the 7B critic remains substantially below SV, confirming that simply adding more candidates to this judge does not close the gap.
>
>     | Method | InfographicVQA | ChartMuseum | ChartQAPro |
>     | --- | --- | --- | --- |
>     | LLaVA-Critic-7B (pointwise) | 82.3 | 34.6 | 56.5 |
>     | **SV w/ GPT-4o Verdict** | **88.3** | **49.3** | **63.4** |
> 4. **Results on LLaVA-Critic-72B judging 3 draft experts:**
>
>     We further repeat the (2) experiment with LLaVA-Critic-72B as a stronger judge (on 1000 random examples per dataset). LLaVA-Critic-72B is still notably weaker than SV on all three benchmarks and its judge-side token usage is on a similar or higher scale than the 7B case, **so it does not offer a great cost–performance trade-off either.**
>
>     | Method | InfographicVQA | ChartMuseum | ChartQAPro |
>     | --- | --- | --- | --- |
>     | LLaVA-Critic-72B (pointwise) | 82.6 | 40.4 | 53.5 |
>     | LLaVA-Critic-72B (pairwise) | 83.1 | 43.0 | 55.8 |
>     | **SV w/ GPT-4o Verdict** | **88.3** | **49.3** | **63.4** |
>
> Under matched prefill and comparable judge/verdict token costs, SV consistently outperforms standard LMM-as-a-Judge frameworks (LLaVA-Critic 7B and 72B). This demonstrates that SV provides a more effective and efficient way than selection-based judging for information-intensive visual reasoning.
>
> [1] LLaVA-Critic: Learning to Evaluate Multimodal Models, CVPR 2025

---

> ### Author Response · Authors · 2025-11-21
> **Response to Reviewer 65ys (Part 7/7)**
>
> ### **Q3: Handling long reasoning trajectories**
>
> We appreciate this thoughtful question about potential context length issues. This occurs rarely and does not impact SV's effectiveness.
>
> In practice, we do occasionally observe very long reasoning trajectories from GLM-4.1V-Thinking-9B and MiMO-VL-RL-7B. But this only happens in 1.5% and 0.5% of their generation, typically when models enter repetitive verification loops after having already completed the essential reasoning. To handle this, we apply standard max-token truncation during reasoning trajectory generation, which preserves the informative early part of the trajectory. After truncation, the combined verdict input remains well below the verdict model’s context limit in all experiments, so the verdict never encounters a length overflow.
>
>
>
> ### **Q4: Formatting (Table 7 → Figure 7)**
>
> We thank the reviewer for pointing this out. We have corrected it in the revised manuscript accordingly.

---

> > ### Comment · Area_Chair_ECnx · 2025-11-26
> >
> > Dear Reviewer,
> >
> > Thanks for your time and effort in reviewing ICLR2026 submissions. The authors have provided their responses to your reviews. Please read and raise your further comments, and discuss with the authors.
> >
> > Best regards,
> >
> > Your AC

---

> > ### Comment · Reviewer_65ys · 2025-11-26
> >
> > I would like to thank the authors for their detailed responses addressing my questions. The additional analyses have significantly strengthened the paper and enhanced my understanding of the proposed framework. I will raise my score accordingly.
> >
> > Regarding W3.3: I have concerns about the fairness of comparing LLaVA-Critic-7B/72B against GPT-4o as the Verdict model. Given that LLaVA-Critic was trained on a limited set of open-source samples while GPT-4o remains a black-box proprietary model, this comparison may not provide meaningful insights into the framework's effectiveness. I would recommend comparing against open-source models (e.g., the base model LLaVA-OneVision 7B) serving as the Verdict to enable a more equitable and interpretable evaluation.

---

> ### Author Response · Authors · 2025-11-27
> **Response to Reviewer 65ys**
>
> ### **Follow-up on W3.3: Fair LMM-as-a-Judge Comparison**
>
> Thank you very much for the positive feedback and for raising this important point about the fairness of the comparison in W3.3. We agree that a more equitable and interpretable evaluation is to compare LMM-as-a-Judge methods under a shared open-source backbone. To this end, we conducted additional experiments where **we used LLaVA-OneVision-7B as the verdict model in SV, and compared it against LLaVA-Critic-7B/72B** (both pointwise and pairwise variants). In all settings, the judge/verdict receives the same three draft experts’ reasoning paths as input, ensuring a fair comparison. We have incorporated these new results and discussion into Appendix G of the revised manuscript.
>
> The results in Tables 1 and 2 show that **SV with a LLaVA-OneVision-7B verdict consistently outperforms all LLaVA-Critic variants across benchmarks.** Concretely, SV improves over LLaVA-Critic-7B by 0.5–6.6%, and even surpasses LLaVA-Critic-72B by 0.5-2%, despite using only a 7B verdict. This demonstrates that the advantage primarily stems from SV’s framework rather than from simply using a stronger verdict model.
>
> In terms of efficiency (Table 3), SV uses fewer or comparable tokens than LLaVA-Critic-7B while achieving higher performance, demonstrating **a better cost–performance trade-off**.
>
> Table 1: Performance of SV w/ LLaVA-OneVision-7B verdict vs. LLaVA-Critic-7B as judge.
>
> | Method | InfographicVQA | ChartMuseum | ChartQAPro |
> | --- | --- | --- | --- |
> | LLaVA-Critic-7B (pointwise) | 83.5 | 40.1 | 52.4 |
> | LLaVA-Critic-7B (pairwise) | 81.4 | 38.9 | 52.1 |
> | **SV w/ LLaVA-OneVision-7B verdict** | **84.0** | **44.1** | **59.0** |
>
> Table 2: Performance of SV w/ LLaVA-OneVision-7B verdict vs. LLaVA-Critic-72B as judge on 1000 randomly sampled instances.
>
> | Method | InfographicVQA | ChartMuseum | ChartQAPro |
> | --- | --- | --- | --- |
> | LLaVA-Critic-72B (pointwise) | 82.6 | 40.4 | 53.5 |
> | LLaVA-Critic-72B (pairwise) | 83.1 | 43.0 | 55.8 |
> | **SV w/ LLaVA-OneVision-7B verdict** | **83.6** | **44.1** | **57.8** |
>
> Table 3: Average judge/verdict tokens per sample, including input and output tokens.
>
> | Method | InfographicVQA | ChartMuseum | ChartQAPro |
> | --- | --- | --- | --- |
> | LLaVA-Critic-7B (pointwise) | 1053.3 | 3723.7 | 1290.7 |
> | LLaVA-Critic-7B (pairwise) | 1342.9 | 4689.4 | 1759.1 |
> | **SV w/ LLaVA-OneVision-7B verdict** | **702.4** | **3123.0** | **1442.4** |

---

### Author Response · Authors · 2025-11-30
**Final Remarks from Authors**

For clarity, we refer to Reviewers 65ys, Cnwz, 656R, and E8fc as **R1, R2, R3, and R4**, respectively.

We sincerely thank all reviewers for their thoughtful and constructive feedback and their active engagement during the discussion period. We note that **all three reviewers who engaged in the discussion (R1, R3, R4) indicated that their main concerns had been addressed and** **raised their scores before November 27 EST** (the widespread leakage incident), leading to improved scores of **6, 6, 8, 6** (from 4, 6, 6, 4) with their original confidence levels unchanged. They explicitly indicated their satisfaction and score increases in their timestamped comments below. We hope this context, together with the summary below, will be helpful.

We have incorporated suggested experiments, discussions, and analyses into the revised manuscript, with revisions highlighted in blue. Below, we summarize our core contributions and major updates introduced during the rebuttal.

---

### **Core contributions of Speculative Verdict (SV)**

We thank the reviewers for emphasizing that solving information-intensive visual reasoning is critical *(highlighted by all reviewers)*, and for recognizing our work’s main contributions:

1. **Novel training-free framework** *(highlighted by R2, R3):* Inspired by the high-level principle of speculative decoding, we address the core challenges of information-intensive images by proposing SV, a training-free framework that combines diverse small VLM experts with a large verdict model to verify and synthesize their reasoning.
2. **Strong empirical performance** *(highlighted by R1, R2, R3):* SV consistently outperforms strong large models and tool-driven methods, achieving 47-53% error correction and generalizing to diverse visual reasoning tasks.
3. **Superior cost-efficiency** *(highlighted by R2, R3):* SV delivers substantially better cost–performance trade-offs. SV achieves or even surpasses o1-level performance while using only 15–26% of its cost, underscoring its practical value for deployment.
4. **Comprehensive analysis** *(highlighted by R3, R4):* We provide extensive empirical ablations and analyses of consensus selection and draft expert complementarity, revealing how SV exploits complementary reasoning strengths across experts to synthesize correct answers.

---

### **Major updates of experimental results during rebuttal**

- `Section 4.6`: Cost-efficiency analysis comparing SV to large model reasoning baselines, including GPT-4o and o1 *(addressing R2’s W2, R3’s W2 & Q2, and R4’s W1 & Q1)*.
- `Section 4.5 & Appendix G`: Comparisons with LLaVA-Critic [1], showing SV outperforms the LMM-as-a-Judge paradigm with superior cost-efficiency *(addressing R1’s W3.3 and R3’s W5 & Q5)*.
- `Section 4.2`: Comparison with the additional specialized method Pixel-Reasoner [2] *(addressing R2’s W1)*.
- `Section 4.5 & Appendix C.2`: Add more ablations using small verdicts with token-efficiency analysis *(addressing R1’s W3.2 & Q2 and R4’s W1 & Q1)*.
- `Appendix F`: Ablations on the consensus selection mechanism, showing SV efficiently ensures balanced and diverse expert participation based on cross-model agreement *(addressing R1’s W2)*.
- `Appendix E`: Evaluated SV with alternative draft pools across model architectures and sizes *(addressing R3’s W3 & Q3)*.
- `Section 4.4`: Experimental results on visual reasoning benchmarks TallyQA-Complex [3] and MathVista [4], demonstrating SV’s generalization *(addressing R3’s W6 & Q6)*.

---

### **Major updates of in-depth discussions and analyses during rebuttal**

- `Section 2`: Elaborated on related work covering LMM-as-a-Judge frameworks and general VLM reasoning, strengthening SV’s distinction and positioning *(addressing R1’s W1 and R4’s W2 & Q1)*.
- `Appendix I`: Quantitative analysis of draft expert complementarity *(addressing R1’s W3.1 & Q1)*.
- `Appendix J`: Failure case analysis *(addressing R3’s W4 & Q4)*.

---

We believe these additions comprehensively address the reviewers’ concerns and strengthen our manuscript. We are grateful for the reviewers’ valuable feedback and appreciate the Area Chair’s additional time and effort on behalf of the community in this situation.

[1] LLaVA-Critic: Learning to Evaluate Multimodal Models, CVPR 2025

[2] Pixel Reasoner: Incentivizing Pixel-Space Reasoning with Curiosity-Driven Reinforcement Learning, NeurIPS 2025

[3] TallyQA: Answering Complex Counting Questions, AAAI 2019

[4] MathVista: Evaluating Mathematical Reasoning of Foundation Models in Visual Contexts, ICLR 2024

---

### Meta-Review · Area_Chair_finL · 2025-12-30

**Summary:**

The paper proposes Speculative Verdict (SV) to improve the reasoning ability of a large VLM using small VLMs as draft models. On several benchmarks, the proposed SV achieves better performance compared with several VLM baselines. The main concerns are insufficient analysis and computational cost, but the authors have done a good job during the rebuttal to address them. Although I think the technical novelty is indeed limited and the comparison with Qwen2.5-VL-Instruct on HR-Bench 4K indicates the weakness of SV, the paper should be accepted given the strengths in the empirical study.

**Reviewer Concerns:**

The following are the major concerns that have been addressed during the rebuttal.

1.	Unclear connection to speculative decoding (65ys)
2.	Insufficient motivation and/or analysis (65ys, 656R)
3.	Lack of comparison with baselines such as specialized models and other ensemble methods (Cnwz, 656R)
4.	The computational cost is possibly high (Cnwz, 656R, E8fc)
5.	Table is not clear (656R)
6.	Generalization is unclear (656R)
7.	Literature review missed important papers (E8fc)

The following is outstanding

1.	Novelty is limited (65ys)
2.	Performance of SV is not competitive in some cases (Cnwz)

**Reviewer Scores:**

Reviewer 65ys would increase to 6.

Reviewer Cnwz did not respond but would likely keep 6.

Reviewer 656R would increase to 8.

Reviewer E8fc would increase to 6.

This gives the overall score of 6668.

---

### Decision · Program_Chairs · 2026-01-26

Accept (Poster)